# Select to Think: Unlocking SLM Potential with Local Sufficiency

**Wenxuan Ye** [1 2]   **Yangyang Zhang** [3]   **Xueli An** [2]   **Georg Carle** [1]   **Yunpu Ma** [3 4 5 †]

## Abstract

Small language models (SLMs) offer efficient deployment, yet they often lag behind their larger counterparts (LLMs) in reasoning. Existing remedies either invoke an LLM at points of reasoning divergence, incurring substantial latency and cost, or rely on standard distillation, which is limited by the SLM's capacity to accurately mimic the LLM's complex generative distribution. We address this dilemma by identifying *local sufficiency*: at divergence points, the LLM's preferred token often resides within the SLM's top-$K$ next-token predictions, even when failing to emerge as the SLM top-1 choice. We therefore propose SELECT TO THINK (S2T), which reframes the LLM's role from open-ended generation to selection among the SLM's proposals, simplifying the supervision signal to discrete candidate rankings. Leveraging this, we introduce S2T-LOCAL, which distills the selection logic into the SLM, empowering it to perform autonomous re-ranking without inference-time LLM dependency. Empirically, a 1.5B SLM's top-8 candidates contain the 32B LLM's choice with a 95% hit rate, and S2T-LOCAL improves the 1.5B SLM's Math Avg. over greedy decoding by 24.1% relative gain, matching the efficacy of 8-path self-consistency with single-trajectory efficiency. Code is available at https://github.com/YeRona/Select-to-Think.

## 1. Introduction

Large Language Models (LLMs) demonstrate superior reasoning capabilities, but their high inference costs limit scalable deployment, motivating growing interest in Small Language Models (SLMs) (Schick & Schütze, 2021; Magister et al., 2023; Liu et al., 2024). However, SLMs often underperform on complex multi-step reasoning tasks (Achiam et al., 2023; Yang et al., 2024). To navigate this trade-off, collaborative reasoning has emerged as a promising paradigm, leveraging the LLM to assist the SLM during inference (Shnitzer et al., 2023; Kim et al., 2025; Wang et al., 2025b). Early systems primarily rely on query-level routing or cascades, escalating entire prompts to the LLM based on difficulty (Varshney & Baral, 2022; Chen et al., 2024; Ong et al., 2025). Yet, recent analyses reveal that errors are predominantly localized, where a small number of critical tokens can cause irreversible drift from the optimal trajectory (Lin et al., 2025; Fu et al., 2025a). This observation has motivated finer-grained collaboration that intervenes selectively at these divergent points, enabling sparse LLM interventions (Huang et al., 2026).

Despite the sparsity of interventions, these approaches remain constrained by latency and computational cost, as each correction necessitates a synchronous LLM call with substantial prefill overhead. Alternatively, attempting to internalize this reasoning capability via standard distillation is hindered by the inherent *capacity gap* (Li et al., 2025). Forcing the SLM to match the teacher's distribution over a *vast, open-ended vocabulary* often compels it to mimic probability peaks that contradict its internal priors, resulting in brittle imitation rather than robust decision rules (Cho & Hariharan, 2019; Chen et al., 2025).

To address this dilemma, we shift focus from generative imitation to the potential of the SLM's top-$K$ next-token candidates, raising a key question:

> *At divergence points, does the SLM's candidate set suffice to capture the LLM's preferred choice?*

As illustrated in Figure 1, uncovering such *local sufficiency* implies a fundamental paradigm shift: resource-intensive open-ended LLM generation can be substituted with *selection* over the SLM's own proposals. This reformulation fundamentally alters the learning objective, effectively bypassing the capacity gap: instead of demanding exact mimicry of the teacher's generative peaks, it reduces the challenge to a far simpler goal, identifying the optimal token from a bounded candidate set.

[1]Technical University of Munich [2]Huawei Technologies Duesseldorf GmbH [3]Ludwig Maximilian University of Munich [4]Munich Center for Machine Learning [5]MemAgents Lab. Correspondence to: Wenxuan Ye <wenxuan.ye@tum.de>, Yunpu Ma <cognitive.yunpu@gmail.com>.

*Proceedings of the 43rd International Conference on Machine Learning*, Seoul, South Korea. PMLR 306, 2026. Copyright 2026 by the author(s).

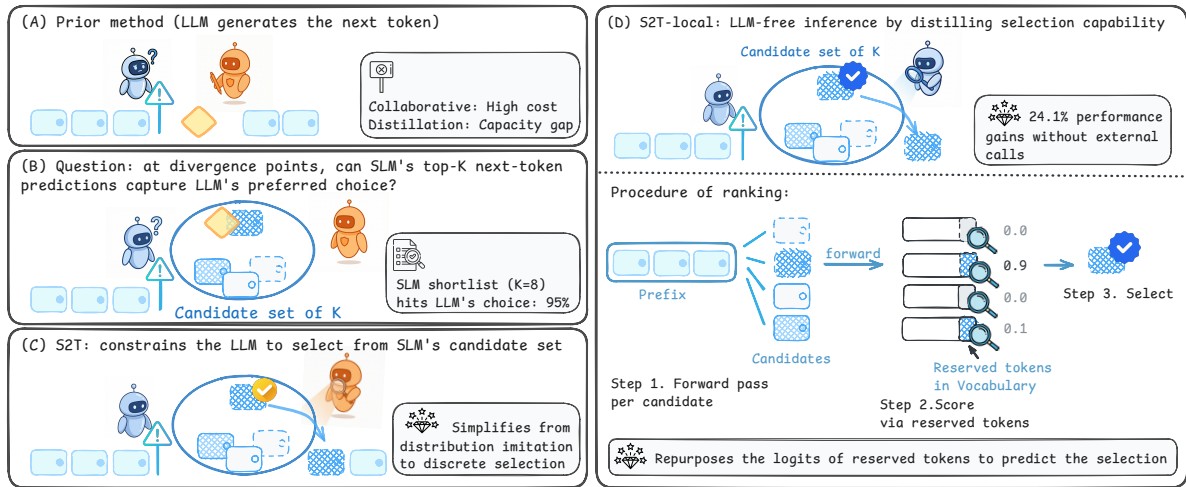

*Figure 1.* The SELECT TO THINK (S2T) Paradigm. *(A) Prior method* faces a dilemma: collaborative inference suffers from high latency due to external calls, while standard distillation is limited by the capacity gap, leading to brittle distribution matching. *(B) Question.* Is LLM generation strictly necessary, or does the SLM's candidate set suffice? Empirically, we observe that the LLM's preferred token resides within the SLM's top-8 predictions in 95% of cases. *(C)* S2T. We reframe LLM guidance as candidate ranking, simplifying distillation from high-dimensional distribution matching to a discrete selection task. *(D)* S2T-LOCAL. We internalize this selection logic by repurposing reserved tokens, yielding an average 24.1% improvement over greedy decoding at comparable cost.

To operationalize this insight, we introduce SELECT TO THINK (S2T). Diverging from prior works that primarily study *when* to intervene, our innovation redefines the intervention *mechanism*. Specifically, we constrain the LLM to *choose* from the SLM's generated options, yielding discrete supervision signals that are inherently easier to distill than complex probability distributions. Capitalizing on these signals, we further propose S2T-LOCAL, which internalizes the selection logic into the SLM itself, creating a teacher-free variant. Since forcing direct probability alignment risks distorting the native distribution, we decouple selection from generation instead. Specifically, adapting the Zero-overhead Inference-time Prediction (ZIP) framework (Manvi et al., 2026), we repurpose the logits of reserved tokens to encode preference scores. This design effectively isolates the selection signal, enabling the SLM to self-evaluate via a learned "inner critic" without perturbing its standard vocabulary statistics.

We systematically evaluate S2T across diverse reasoning benchmarks. *First*, our analysis validates the *local sufficiency*: the 1.5B SLM's top-8 candidates capture the 32B teacher's preference with a 95% hit rate. This phenomenon proves consistent across model scales (down to 0.5B) and architectures. *Second*, we confirm the learnability of this signal, where the internalized selector achieves a robust 67% agreement with the LLM on held-out instances. Translating these factors into performance, S2T-LOCAL improves greedy decoding by an average of 24.1% relative gain at comparable computational cost, effectively match-

ing the performance of 8-path self-consistency with single-trajectory efficiency.

Our contributions are summarized as follows:

- **Insight:** We formalize and empirically verify *local sufficiency*: at divergence points, the LLM's preferred tokens are consistently contained within the SLM's top-K predictions. This finding identifies selection refinement, rather than generative imitation, as a pivotal avenue for SLM optimization.
- **Framework:** SELECT TO THINK (S2T) reframes LLM assistance as explicit selection over SLM proposals, simplifying the distillation objective from complex distribution matching to a tractable ranking task.
- **Method:** We introduce S2T-LOCAL, a teacher-free variant that internalizes the selection logic. By utilizing reserved tokens to encode preference scores, we decouple generation from evaluation, enabling the SLM to perform autonomous re-ranking without external overhead.
- **Experiments:** Through extensive experiments and mechanistic analysis, we demonstrate that S2T-LOCAL improves 1.5B's greedy decoding by an average of 24.1% relative gain, effectively matching the performance of 8-path self-consistency with single-trajectory efficiency.

## 2. Related Works

**Model routing.** To mitigate LLM inference costs, adaptive strategies dynamically delegate generation tasks based on input difficulty. While early systems routed entire

queries (Chen et al., 2024), more recent works enable fine-grained token-level interventions, dynamically invoking the LLM exclusively for critical tokens (Fu et al., 2025a; Huang et al., 2026). However, these approaches still rely on *LLM generation* as the intervention primitive, which requires synchronous teacher calls and injects external tokens into the SLM trajectory.

**Speculative decoding.** Speculative decoding accelerates inference by employing an efficient drafter to propose tokens and a target LLM to verify (Leviathan et al., 2023; Chen et al., 2023). Extensions such as SpecInfer (Miao et al., 2023) and SpecTr (Sun et al., 2023) verify trees or cascades to improve acceptance rates. Recent variants (e.g., SpecReason (Pan et al., 2025)) relax strict token matching by verifying reasoning steps semantically. This line primarily targets latency reduction for LLM inference, and typically retains an external verifier model at test time, whereas our goal is to enhance the SLM's reasoning performance through localized guidance.

**Knowledge distillation.** Standard distillation often requires matching a strong teacher's open-ended output distribution, which can be brittle under large LLM-SLM capacity gaps (Wang et al., 2025a; Zhang et al., 2025). Recent work therefore moves toward selective supervision, for example by emphasizing key reasoning tokens or high-value steps, e.g., RLKD (Peng & Zhang, 2025) and TSD-KD (Kim & Baek, 2026). However, they remain bound to the generative paradigm by optimizing the SLM to match the LLM's distribution over the full vocabulary.

**Test-time compute scaling.** A separate line of work improves reasoning by allocating more inference compute, typically via sampling-and-aggregation (Self-Consistency (Wang et al., 2023)) or explicit search (Tree of Thoughts (Yao et al., 2023)). Recent advances attempt to localize this computation through token-level mechanisms, such as selective latent iterations (Think-at-Hard (Fu et al., 2025b)) or parallel branching (Multiplex Thinking (Tang et al., 2026)). However, these methods generally remain rooted in *optimizing the generative process*, bottlenecked by the model's inherent capacity constraints.

## 3. Preliminaries

### 3.1. Autoregressive decoding

Let $\mathcal{V}$ be a token vocabulary. Given an input $x$ sampled from a data distribution $\mathcal{X}$, an autoregressive model generates a response $y = (y_1, \ldots, y_T)$. Let $y_{<t} := (y_1, \ldots, y_{t-1})$ be the prefix before step $t$, and define the decoding state as $s_t := (x, y_{<t})$. A language model $\pi_\theta$ produces logits $z_t \in \mathbb{R}^{|\mathcal{V}|}$, inducing a next-token distribution $p_\theta(\cdot \mid s_t) = \mathrm{softmax}(z_t)$. Standard inference typically employs greedy

decoding:

$$\hat{y}_t = \arg\max_{v \in \mathcal{V}} p_\theta(v \mid s_t) \tag{1}$$

To model collaborative inference, let $\pi_{\theta_L}$ and $\pi_{\theta_S}$ denote a large teacher (LLM) and a small student (SLM), parameterized by $\theta_L$ and $\theta_S$ respectively. Existing collaborative baselines can be abstracted via a binary trigger variable $g_t \in \{0, 1\}$ that governs the token source:

$$\hat{y}_t = \begin{cases} \arg\max_{v \in \mathcal{V}} p_{\theta_S}(v \mid s_t), & \text{if } g_t = 0, \\ \arg\max_{v \in \mathcal{V}} p_{\theta_L}(v \mid s_t), & \text{if } g_t = 1. \end{cases} \tag{2}$$

For brevity, we present the greedy form, though our formulation generalizes naturally to stochastic sampling. Typically, $g_t$ is determined by a triggering strategy (e.g., uncertainty thresholding).

### 3.2. Zero-overhead Inference-time Prediction (ZIP)

Zero-overhead Inference-time Prediction (ZIP) (Manvi et al., 2026) repurposes a set of reserved tokens $\mathcal{R} \subset \mathcal{V}$, and utilizes their logits to encode auxiliary predictions. These tokens are masked from the generative distribution to preserve valid text generation, yet their logits $z_t[\mathcal{R}]$ remain available to encode auxiliary signals. ZIP computes a scalar prediction $\hat{s}_t$ by applying a readout mapping $\psi : \mathbb{R}^{|\mathcal{R}|} \to \mathbb{R}$ to these reserved logits (e.g., via a weighted linear projection):

$$\hat{s}_t = \psi(z_t[\mathcal{R}]) \tag{3}$$

In this work, we adopt this mechanism to parameterize a lightweight selection score, enabling the model to output a quality score $\hat{s}_t$ conditional on its current state.

## 4. Method: the S2T Framework

We introduce S2T, which reframes the collaborative paradigm by substituting open-ended generation with discrete selection over the SLM's candidates. This shift simplifies the LLM's guidance into a manageable scoring task, which can then be distilled into the SLM to enable high-performance local inference without external LLM calls.

We organize the methodology as follows: Section 4.1 briefly outlines the intervention trigger (determining *when* to intervene). The core of our framework follows: Section 4.2 details the candidate selection protocol (determining *what* and *how* to select), and Section 4.3 describes how this selection logic is internalized for fully local inference. Finally, Section 4.4 analyzes the theoretical advantages of this paradigm.

### 4.1. Intervention trigger

To balance reasoning accuracy with efficiency, we first define an oracle trigger to identify critical decoding steps. Following prior findings that reasoning errors often stem from

a few pivotal tokens (Lin et al., 2025), we utilize the KL divergence $D_{\mathrm{KL}}(p_{\theta_L} \| p_{\theta_S})$ as the metric for disagreement. The binary trigger $g_t$ is activated ($g_t = 1$) if the divergence exceeds the top-$\tau$ percentile observed in the calibration set. To enable local inference, we distill this oracle decision into a lightweight predictor. Given that the trigger module is largely orthogonal to our selection mechanism and has been explored in prior studies (Fu et al., 2025a; Huang et al., 2026), we prioritize our discussion on the selection logic and defer the implementation details of the trigger head to Appendix E.

## 4.2. Selection mechanism

When the trigger activates, S2T replaces standard SLM decoding with *candidate-based selection*. First, the SLM constructs a local candidate set $\mathcal{C}_t$ by identifying the top-$K$ tokens from its next-token distribution (or sampling via multinomial). The decoding step then becomes a maximization over this constrained set, governed by a scoring function $\mathcal{S}(c; s_t)$:

$$\hat{y}_t = \underset{c \in \mathcal{C}_t}{\arg\max} \ \mathcal{S}(c; s_t) \qquad (4)$$

By restricting to student proposals, the choice remains within the student's local support and reduces the LLM's decision space to $K$ candidates.

We instantiate the scoring function $\mathcal{S}(\cdot)$ in two settings, with the algorithmic flow of S2T in Algorithm 1 (Appendix A).

S2T. A collaborative framework where the LLM actively scores the SLM's proposals. The score is defined as the LLM's conditional probability for each candidate: $\mathcal{S}(c; s_t) = p_{\theta_L}(c \mid s_t)$. Here, the LLM provides a restricted supervision signal via ranking the student's proposals, rather than generating open-ended text.

S2T-LOCAL. To remove inference-time dependencies on the LLM, we introduce S2T-LOCAL, a variant that replaces the LLM with a distilled internal selector. Concretely, for each candidate $c \in \mathcal{C}_t$, we construct the augmented state $s_t^c := (x, y_{<t} \oplus c)$. We then perform a forward pass on $s_t^c$ to extract the logits $z_{\theta_S}[\mathcal{R}]$ corresponding to the reserved set $\mathcal{R}$. Following the discrete binning approach, the preference score is computed as:

$$\mathcal{S}_{local}(c; s_t) = \mathbf{v}^\top \mathrm{softmax}\left(z_{\theta_S}(s_t^c)[\mathcal{R}]\right) \qquad (5)$$

where $\mathbf{v}$ is a vector of predefined bin values. In practice, we maximize efficiency by batching the $K$ augmented inputs $\{s_t^c\}_{c \in \mathcal{C}_t}$ to compute all candidate scores in a single parallel forward pass. Finally, we append the highest-scoring candidate to the prefix and continue decoding from the updated state.

## 4.3. Distillation of the selection logic

**Data Collection.** We construct a training dataset $\mathcal{X}$ by rolling out the SLM. At each intervention step, we generate the local candidate set $\mathcal{C}_t$ from the student's next-token distribution $p_{\theta_S}(\cdot \mid s_t)$. The teacher then annotates these candidates by evaluating their next-token likelihood under the same prefix $s_t$: $p_{\theta_L}(c \mid s_t)$ for each $c \in \mathcal{C}_t$. The optimal candidate $c^\star$ is then identified as the one with the highest likelihood among the set.

**Optimization objective.** We jointly minimize a selection loss and a stability regularizer:

$$\mathcal{L}(\theta_S) = \mathcal{L}_{\mathrm{sel}} + \beta \cdot \mathcal{L}_{\mathrm{reg}} \qquad (6)$$

The selection term $\mathcal{L}_{\mathrm{sel}}$ maximizes the likelihood of the target candidate $c^\star$ over the set $\mathcal{C}_t$. We formulate this as a cross-entropy loss over the scores predicted by the student:

$$\mathcal{L}_{\mathrm{sel}} = -\log \frac{\exp\left(\mathcal{S}_{\mathrm{local}}(c^\star; s_t)/T\right)}{\sum_{c' \in \mathcal{C}_t} \exp\left(\mathcal{S}_{\mathrm{local}}(c'; s_t)/T\right)} \qquad (7)$$

where $T$ is a temperature hyperparameter controlling the sharpness of the distribution. Minimizing this term increases the discriminative margin between the target $c^\star$ and other candidates, effectively aligning the reserved-token readout to correctly capture the teacher's preference.

To prevent the fine-tuning from distorting the model's general linguistic capabilities, we enforce a stability constraint $\mathcal{L}_{\mathrm{reg}}$. We calculate the KL divergence between the fine-tuned distribution $\pi_{\theta_S}$ and the frozen base model $\pi_{S,\mathrm{base}}$ over the standard text vocabulary $\mathcal{V}_{\mathrm{text}} = \mathcal{V} \setminus \mathcal{R}$:

$$\mathcal{L}_{\mathrm{reg}} = \mathbb{D}_{\mathrm{KL}}\left(\pi_{S,\mathrm{base}}(\cdot \mid s_t) \,\big\|\, \pi_{\theta_S}(\cdot \mid s_t)\right)_{\mathcal{V}_{\mathrm{text}}} \qquad (8)$$

This ensures that while the reserved tokens $\mathcal{R}$ learn to carry discriminative signals, the standard tokens continue to function as a coherent language model.

## 4.4. Analysis of the collaborative selection mechanism

Following standard formulations in imitation learning (Ross et al., 2011; Sutton & Barto, 1998), we view autoregressive decoding as a horizon-$T$ sequential decision process. Given parameters $\theta$, the decoder induces a decision rule $f_\theta : \mathcal{X} \times \mathcal{V}^\star \to \mathcal{V}$, where $\mathcal{V}^\star = \bigcup_{n=0}^{\infty} \mathcal{V}^n$ denotes the set of all possible prefixes, and $f_\theta$ represents the decoding mechanism.

In S2T framework, the decoder first determines a dynamic candidate size $k_t$ based on the trigger $g_t(s_t) \in \{0, 1\}$:

$$k_t = K \ \text{if} \ g_t(s_t) = 1 \ \text{else} \ 1 \qquad (9)$$

The search space is then constrained to a local candidate set $\mathcal{C}_t$ comprising the $k_t$ tokens from the SLM's probability

distribution. Finally, the decision rule $f_\theta$ selects the next token via the scoring function $\mathcal{S}$, as in Equation (4). This unified formulation recovers standard decoding as a special case: when $g_t = 0$, the set $\mathcal{C}_t$ collapses to a single token, forcing $f_\theta$ to follow the SLM's original path regardless of the scoring function.

We assume access to a teacher *reference model* that induces an optimal reference token $y^\star(s_t) \in \mathcal{V}$ at each step (e.g., $y^\star(s_t) = \arg\max_{v \in \mathcal{V}} p_{\theta_L}(v \mid s_t)$). We measure the instantaneous 0-1 decoding error as:

$$\ell(s_t, \hat{y}_t) := \mathbb{I}[\hat{y}_t \neq y^\star(s_t)] \in \{0, 1\} \qquad (10)$$

Consequently, for any state $s_t$, the error can be decomposed into two distinct sources:

$$\mathbb{I}[f_\theta(s_t) \neq y^\star(s_t)] = \underbrace{\mathbb{I}[y^\star(s_t) \notin \mathcal{C}_t]}_{\delta_{\text{hit}}^{(t)}(k_t; s_t)}$$
$$+ \underbrace{\mathbb{I}[y^\star(s_t) \in \mathcal{C}_t, \, f_\theta(s_t) \neq y^\star(s_t)]}_{\delta_{\text{sel}}^{(t)}(k_t; s_t)}$$
$$(11)$$

where $\delta_{\text{hit}}$ and $\delta_{\text{sel}}$ denote the hit failure and selection failure, respectively.

**Proposition.** *Let $d_\theta^t$ denote the state distribution at step $t$ induced by rolling out $f_\theta$. For any model $\theta$ and candidate size $K$, the cumulative error decomposes as*

$$\mathcal{E}(\theta) = \sum_{t=1}^{T} \mathbb{E}_{s_t \sim d_\theta^t}\Big[\ell(s_t, f_\theta(s_t))\Big]$$
$$= \sum_{t=1}^{T} \mathbb{E}_{s_t \sim d_\theta^t}\Big[(1 - g_t(s_t)) \cdot \delta_{\text{hit}}^{(t)}(1; s_t) + \qquad (12)$$
$$g_t(s_t) \cdot \Big(\delta_{\text{hit}}^{(t)}(K; s_t) + \delta_{\text{sel}}^{(t)}(K; s_t)\Big)\Big]$$

This proposition formalizes the trade-off between search coverage and selection precision. While increasing $K$ reduces hit failure ($\delta_{\text{hit}}$) by expanding the search space, it introduces selection error ($\delta_{\text{sel}}$) due to increased discriminative difficulty. S2T reduces cumulative error by balancing increased candidate coverage (lower $\delta_{\text{hit}}$ at moderate $K$) against minimal selection difficulty ($\delta_{\text{sel}}$).

## 5. Experiments

To empirically validate S2T, we conduct a comprehensive evaluation designed to answer four research questions:

- RQ1 (Overall performance): How does S2T compare to LLM-free and collaborative baselines?
- RQ2 (Paradigm validity): Does the local sufficiency hypothesis hold, and can the *selection* paradigm recover the performance of open-ended *generation*?

- RQ3 (Distillation efficacy): Can the SLM effectively internalize teacher preferences to enable autonomous scoring mechanism?
- RQ4 (Ablation): How do key design choices impact the overall performance?

### 5.1. Setup

We outline key configurations below, with full implementation details provided in Appendix C.

**Models and Datasets.** We utilize the Qwen2.5-Instruct family (Qwen, 2025), pairing 0.5B and 1.5B students with a 32B teacher to simulate realistic capacity gaps. We evaluate on mathematical reasoning (GSM8K (Cobbe et al., 2021), MATH (Lightman et al., 2024), OlympiadBench (He et al., 2024), AIME25 (Zhang & Math-AI, 2025)), coding (HumanEval (Chen et al., 2021)), and broad knowledge (MMLU-Pro (Wang et al., 2024)). We adopt a zero-shot protocol with a task-specific system instruction.

**S2T Configurations.** Using KL divergence as the intervention signal, S2T directs the SLM to sample $K$ candidates via multinomial and LLM to rank. To enable local inference, we distill this ranking logic into the SLM. Coupled with a learned gating mechanism to trigger intervention, S2T-LOCAL empowers the SLM to achieve fully autonomous inference without external LLM calls. Unless otherwise specified, we set the candidate size $K = 8$ and the intervention budget to $\tau = 1\%$. Results are reported as average metrics per instance, across three random seeds to ensure statistical robustness.

To rigorously evaluate out-of-distribution (OOD) generalization, we train S2T *exclusively* on the MATH training split, leaving all other benchmarks entirely unseen. We curate a dataset of 2k trajectories by selecting the top $\tau = 10\%$ decoding steps by KL divergence, with $K = 16$ candidates per step annotated by the teacher. For optimization, we apply LoRA adapters on attention and MLP modules, while additionally unfreezing the specific $\text{lm}_{\text{head}}$ embeddings for the reserved tokens $\mathcal{R}$. Specifically, we use $|\mathcal{R}| = 16$ for Qwen2.5-1.5B and $|\mathcal{R}| = 12$ for Qwen2.5-0.5B in the main experiments. For each setting, the predefined bin vector $v$ is fixed and non-learnable, with uniformly spaced anchors over $[0, 1]$, i.e., $v_i = (i - 1)/(|\mathcal{R}| - 1)$, to convert the predicted bin distribution into the continuous preference score in Eq. 5. This configuration yields highly efficient adaptation: 35.7M and 74.3M trainable parameters for the 0.5B and 1.5B models, respectively ($< 6\%$ of total capacity).

**Baselines.** We categorize baselines by their reliance on external LLMs during inference. *SLM-only* methods operate independently, including *SLM Greedy*, the standard distillation baseline *DistilQwen2.5* (Wang et al., 2025a), and *TSD-KD* (Kim & Baek, 2026), which applies distillation

*Table 1. Main results on mathematical and general reasoning benchmarks.* (1) *Local Efficacy:* S2T-LOCAL significantly outperforms distillation mechanisms and adaptive compute methods, and demonstrates strong OOD generalization. Notably, it matches the accuracy of 8-path self-consistency (Maj@8) with single-trajectory efficiency. (2) *Collaborative Performance:* S2T matches the generative methods.

| Param. | Benchmark | SLM-only | | | | | | | | LLM-involved | | | |
|---|---|---|---|---|---|---|---|---|---|---|---|---|---|
| | | Greedy | Sample | Distil | Maj@8 | TSD-KD | Mul-T | TaH | S2T-LOCAL | R2R | SpecR | Takeover | S2T |
| | GSM8k | 44.5 | 48.0 | 44.5 | 59.1 | 46.4 | 46.5 | 48.0 | **61.6** | 85.7 | **87.3** | 87.0 | 83.2 |
| | MATH500 | 30.4 | 32.7 | 34.0 | 42.2 | 33.0 | 33.0 | 38.7 | **46.8** | **74.4** | 72.2 | 69.7 | 72.9 |
| | OlympiadBench | 5.8 | 5.5 | 8.1 | **10.8** | 7.7 | 5.8 | 7.0 | 8.5 | 24.9 | **27.4** | 25.4 | 26.8 |
| 0.5B | AIME25 | 0.0 | 0.0 | 0.0 | 0.0 | 0.0 | 0.0 | 0.0 | 0.0 | 4.4 | 3.3 | 4.4 | **5.5** |
| | *Math Avg.* | 20.2 | 21.6 | 21.7 | 28.0 | 21.8 | 21.3 | 23.4 | **29.2** | 47.4 | **47.6** | 46.6 | 47.1 |
| | HumanEval | 18.9 | 16.4 | 22.9 | - | 23.0 | 19.7 | 20.3 | **26.7** | **64.6** | 62.4 | 62.0 | 61.0 |
| | MMLU-Pro | 11.3 | 11.4 | 14.3 | **20.0** | 11.4 | 12.0 | 13.4 | 18.7 | 33.7 | 35.8 | 33.7 | **36.9** |
| | GSM8k | 72.1 | 75.0 | 79.1 | 81.4 | 74.5 | 76.0 | 78.4 | **86.6** | **96.4** | 83.0 | 94.9 | 95.6 |
| | MATH500 | 54.4 | 55.4 | 60.4 | 64.1 | 58.1 | 58.5 | 60.9 | **67.5** | 78.3 | 73.6 | 80.0 | **81.1** |
| | OlympiadBench | 20.1 | 22.1 | 20.0 | 26.1 | 19.6 | 22.8 | 17.1 | **27.0** | 42.5 | **42.6** | 40.1 | 37.5 |
| 1.5B | AIME25 | 1.1 | 1.1 | 1.1 | **3.3** | 1.1 | 2.2 | 2.2 | 2.2 | 8.8 | **10.0** | 7.7 | 8.8 |
| | *Math Avg.* | 36.9 | 38.4 | 40.2 | 43.7 | 38.3 | 39.9 | 39.7 | **45.8** | 56.5 | 52.3 | 55.7 | 55.8 |
| | HumanEval | 42.7 | 39.2 | 48.1 | - | 47.8 | 44.7 | 43.6 | **56.9** | 72.2 | 68.0 | **73.2** | 72.6 |
| | MMLU-Pro | 22.9 | 23.6 | 27.9 | 28.6 | 23.2 | 24.5 | 21.9 | **33.6** | 56.1 | **62.4** | 57.0 | 55.6 |

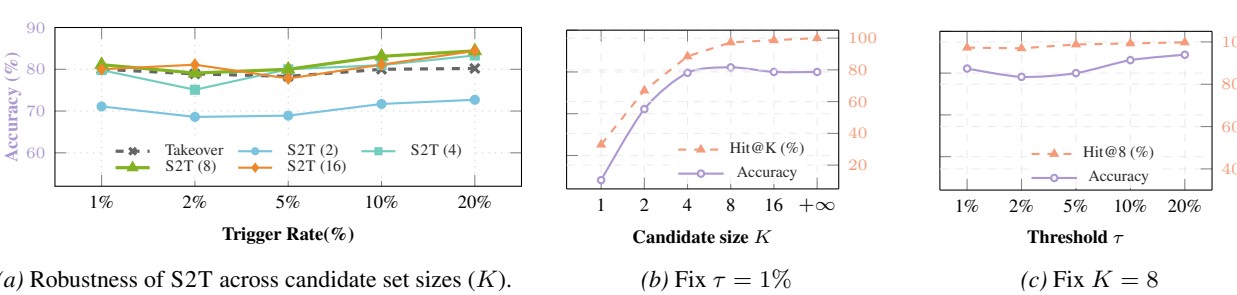

*(a) Robustness of S2T across candidate set sizes ($K$).*     *(b) Fix $\tau = 1\%$*     *(c) Fix $K = 8$*

*Figure 2. Analysis of* S2T *performance (Qwen2.5, 1.5B SLM on MATH).* (a) Accuracy improves consistently as the candidate size $K$ and trigger ratio $\tau$ increase; notably, S2T matches generative baselines even at a moderate $K = 8$. (b) Effect of $K$: Performance saturates beyond $K = 8$. (c) Effect of $\tau$: Accuracy improves via more frequent interventions, while the hit rate remains stable.

exclusively to critical tokens. We also include test-time scaling strategies: *Self-Consistency* (Maj@8) (Wang et al., 2023) for ensemble sampling, *Multiplex-Thinking* (Mul-T) (Tang et al., 2026) via soft token aggregation, and *Think-at-Hard* (TaH) (Fu et al., 2025b) for dynamic budget allocation. Collaborative methods include generative collaboration frameworks like *SpecReason* (SpecR) (Pan et al., 2025) and *R2R* (Fu et al., 2025a). Moreover, to isolate the benefit of selection versus generation, we implement *Takeover*, a controlled baseline that uses the same trigger schedule and budget as S2T but instructs the LLM to *generate* the next token rather than perform selection.

**Metric.** Hit rate (Hit@$K$) measures the probability that the LLM's preferred token is included within the SLM's top-$K$ candidate set, complementary to $\delta_{\text{hit}}$. Agree@1 denotes the probability that the distilled selector successfully identifies the LLM's choice from the candidates.

## 5.2. RQ1: Overall performance

Table 1 presents a comprehensive evaluation across two model scales (0.5B and 1.5B) on six diverse benchmarks.

**RQ1.1: How effective is S2T-LOCAL in math reasoning?** It achieves significant improvements compared to both distillation baselines and recent adaptive strategies grounded in non-uniform compute allocation for "hard" tokens. We observe *a 24.1% relative improvement on the 1.5B model's Math Avg. score*, increasing from 36.9 to 45.8; this gain is even larger at the 0.5B scale, where S2T-LOCAL *improves Math Avg. from 20.2 to 29.2, corresponding to a 44.6% relative gain*. Remarkably, the single-trajectory S2T-LOCAL matches Self-consistency (Maj@8) with about $1/8$ of the compute cost.

This performance leap is fundamentally rooted in the high hit rate across model scales. As demonstrated in our subsequent analysis, while the 1.5B model achieves a Hit@8 of 95%, even the smaller 0.5B SLM reaches a striking 83%.

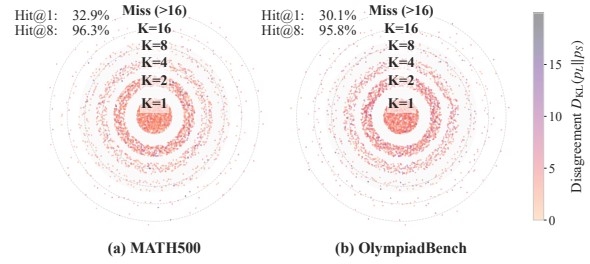

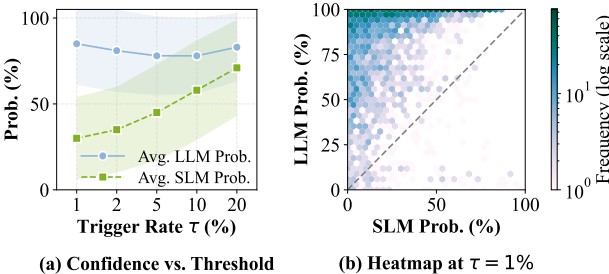

*Figure 3. Validation of local sufficiency.* Radial visualization of intervention steps, where radial distance indicates the minimal $K$ needed to capture the LLM's choice, and color denotes KL divergence. The dense central clustering underscores the significant coverage gain: while Hit@1 is limited to only 30%, a compact candidate set of $K = 8$ successfully captures the target in over **95%** of cases. It confirms that LLM-preferred tokens reside in the SLM's candidate set yet are overlooked by greedy decoding.

*Figure 4. Analysis of probability gaps.* (a) Statistics across $\tau$: lower thresholds ($\tau$) strictly target severe mismatches (high $p_{\theta_L}$, low $p_{\theta_S}$), while increasing $\tau$ incorporates milder disagreements. (b) Heatmap at $\tau = 1\%$ (dashed: $p_{\theta_L} = p_{\theta_S}$): the top-left mass highlights a "rescue" regime, demonstrating how the method exploits local sufficiency by recovering teacher-endorsed tokens that are suppressed by the student.

This suggests that the optimal reasoning path often resides within the model's top-$K$ proposals; by effectively exploiting this hidden information, S2T-LOCAL converts this untapped potential into substantial performance gain.

**RQ1.2: How generalizable is S2T-LOCAL?** Beyond math tasks, S2T-LOCAL demonstrates exceptional transferability to diverse domains, including code generation (HumanEval) and complex general reasoning (MMLU-Pro). As shown in Table 1, we observe consistent performance gains across these benchmarks, confirming that the learned selection mechanism captures broadly transferable reasoning preferences rather than relying on domain-specific heuristics. We further evaluate the same MATH-trained checkpoints on additional non-math domains, including factuality, science question answering, and open-ended instruction following; the results are summarized in Table A5.

**RQ1.3: How effective is selection-based guidance in collaborative settings?** In the collaborative setting, S2T is consistently comparable to strong collaborative methods across diverse benchmarks, as demonstrated in Table 1. In addition, we evaluate collaboration performance by varying the candidate size ($K$) and the intervention rate ($\tau$). Figure 2a shows consistent gains with increasing $K$ or $\tau$, quickly saturating at larger budgets. Crucially, selection-based guidance ($K \geq 8$) matches the generative baseline, demonstrating that discriminative selection is sufficient to recover LLM reasoning without the need for expensive generation. (See Table A3 for results on additional benchmarks.)

### 5.3. RQ2: Paradigm validity

A core hypothesis underlying S2T is that the SLM retains the underlying capacity to match the LLM's reasoning, yet this ability is often masked by the sub-optimality of standard decoding strategies. In this section, we empirically validate this premise. For the following analysis, we default to the 1.5B SLM and MATH500, unless noted otherwise.

**RQ2.1: Does the candidate set cover the token preferred by LLM?** In Figure 2b, fixing the intervention frequency at $\tau = 1\%$ and varying candidate size $K$, we observe that both hit rate and downstream accuracy rise rapidly and quickly saturate, exhibiting diminishing returns beyond small values of $K = 8$. Conversely, with fixed $K = 8$ (Figure 2c), the hit rate remains stable, while accuracy improves as higher $\tau$ enables more frequent refinements.

*Validation of local sufficiency.* We visualize the candidate hit rate in Figure 3. While greedy decoding (Hit@1) aligns with the LLM only 30% of the time, expanding to a modest $K = 8$ captures the target token in over 95% of cases. This empirical evidence confirms that the LLM's preferred tokens almost always lie within the SLM's local distribution. Consequently, failures at critical steps stem not from an absence of candidates, but from an inability to prioritize the available candidates.

*Generalization.* We further substantiate this observation across the 0.5B parameter scale, diverse model architectures (Llama-3 (Grattafiori et al., 2024), Gemma-2 (Team et al., 2024), Phi-3 (Abdin et al., 2024)), and general reasoning tasks (see Figures A1 and A2 and Section F.5). Surprisingly, the 0.5B SLM achieves a Hit@8 exceeding 83%, despite its significantly smaller parameter count. The consistent high coverage across different scales and domains confirms that local sufficiency is an inherent characteristic, providing a robust foundation for selection-based inference.

**RQ2.2: How does the selection mechanism utilize this potential?** In Figure 4(a), lower thresholds maximize probability divergence, suggesting that S2T prioritizes tokens where SLM uncertainty contrasts most sharply with teacher preference. As visualized in Figure 4(b), the top-left concentration highlights the mechanism's ability to recover LLM-endorsed tokens from the candidate set, even when

*Table 2. Selector alignment.* The distilled selector achieves high alignment with the LLM across benchmarks. Crucially, it effectively *recovers* LLM-preferred tokens that are severely *underestimated* by the SLM. (OB: OlympiadBench).

| Metric | Benchmark | | | Random |
|---|---|---|---|---|
| | **MATH500** | **OB** | **MMLU-Pro** | |
| Agree@1 ($\uparrow$) | 69.4% | 71.4% | 67.2% | 12.5% |
| Spearman $\rho$ ($\uparrow$) | 0.63 | 0.65 | 0.61 | 0.00 |
| SLM on sel. | 28.5% | 28.4% | 25.2% | - |
| LLM on sel. | 69.9% | 67.0% | 45.4% | - |
| Score on sel. | 84.2% | 80.3% | 77.3% | - |

*Table 3. Efficiency analysis per instance.* S2T-LOCAL maintains standard SLM inference speeds, reducing latency by $\sim$75% compared to collaborative methods.

| Method | MATH | | OlympiadBench | | AIME25 | |
|---|---|---|---|---|---|---|
| | **Tokens** | **Time (s)** | **Tokens** | **Time (s)** | **Tokens** | **Time (s)** |
| SLM | 493.8 | 9.3 | 827.5 | 15.5 | 1631.6 | 30.7 |
| LLM | 534.6 | 31.9 | 647.2 | 38.5 | 1229.3 | 73.7 |
| Takeover | 518.1 | 40.5 | 938.8 | 74.7 | 1028.7 | 81.5 |
| S2T | 513.3 | 40.5 | 870.8 | 69.7 | 1079.8 | 85.8 |
| S2T-LOCAL | 512.4 | 10.2 | 903.3 | 18.0 | 1053.8 | 21.0 |
| *vs. Takeover* | -1.1% | -74.8% | -3.8% | -75.9% | +2.4% | -74.2% |

they are severely underestimated by greedy decoding. By leveraging the LLM's superior performance to minimize selection error, S2T effectively capitalizes on the high inherent hit rate, successfully translating this potential into superior task accuracy as analyzed in Section 4.4.

**RQ2.3: How does hit rate evolve as the candidate set size $K$ increases?** The rank CDFs across various tasks (Figure A3) consistently exhibit a steep initial surge in Hit@$K$. While the baseline accuracy varies, we observe a universal saturation point around $K = 8$, beyond which marginal gains rapidly diminish.

### 5.4. RQ3: Distillation efficacy

**RQ3.1: How accurately does the distilled selector replicate the teacher's preferences?** As shown in Table 2, the selector achieves a robust Agree@1 of 68%. This high alignment proves that the student has successfully internalized the teacher's judgment, directly driving the performance gains observed in Table 1. In addition, confidence statistics reveal a similar corrective dynamic as in Figure 4: the selector typically rescues tokens with low SLM probability but high LLM certainty, effectively recovering valid reasoning paths that were underestimated by the greedy decoding. To further examine the alignment between predicted scores and teacher ranks, we present Figure A5, which reveals a "winner-takes-all" characteristic where the selector decisively isolates the top candidate while suppressing sub-optimal ones.

Beyond the Qwen family, the learned selection behavior

also transfers to Llama-3.x: using Llama-3.2-1B-Instruct as the SLM and Llama-3.1-8B-Instruct as the LLM teacher, the full S2T-LOCAL pipeline consistently improves over greedy decoding across evaluated benchmarks; detailed results are provided in Section F.4.

**RQ3.2: Can the selection capability generalize to OOD tasks?** Remarkably, despite being trained exclusively on the MATH dataset, the distilled selector demonstrates robust generalization, achieving high accuracy across OOD benchmarks, as shown in Table 1. See Table A9 for a comprehensive analysis across 0.5B models and additional tasks.

**RQ3.3: What is the computational overhead of S2T-LOCAL?** The inference process augments standard SLM decoding with a lightweight projection for trigger evaluation. Upon triggering, the SLM generates and scores the candidate set $\mathcal{C}_t$; thus, the additional computational cost is dominated by these sparse interventions. As quantified by the average token count and wall-clock time in Table 3, S2T-LOCAL achieves substantial speedups, validating its efficiency against intensive baselines. Detailed comparisons are provided in Table A13 and Table A14.

### 5.5. RQ4: Ablation

**RQ4.1: Is local sufficiency tied to a particular trigger metric?** Table A10 shows that local sufficiency is robust across different trigger metrics, but downstream accuracy still depends on trigger quality. Across KL divergence, SLM entropy, and random triggers, Hit@8 remains consistently high, indicating that the LLM-preferred token is usually contained in the SLM's local candidate set even when the exact intervention locations change. However, S2T accuracy varies substantially across trigger metrics: KL divergence yields the strongest performance as it more effectively identifies positions where selection can correct student failures. Thus, the trigger metric mainly determines *where* selection is useful, while the high Hit@8 across metrics supports the broader local-sufficiency hypothesis.

**RQ4.2: How does candidate generation strategy impact ranking efficacy?** As shown in Table A15, candidate generation strongly affects the effectiveness of selection. Common sampling settings, such as top-$p = 0.7$, truncate the SLM's local probability tail and reduce the candidate set to $|\mathcal{C}_t| \approx 1.5$ on average, often removing teacher-preferred tokens before the selector can rank them. In contrast, S2T uses $p = 1.0$ only at trigger points to preserve the SLM's full local support when constructing the bounded candidate pool with $K = 8$, improving accuracy to $81.1\%$. The decoding procedure remains controlled: the selector deterministically chooses from this bounded candidate pool, and all non-trigger steps remain standard greedy decoding. Thus, S2T mitigates the usual concerns associated with fully stochastic decoding, while preserving the local support

needed for teacher-aligned selection.

**RQ4.3: What are the key training strategies for an effective distilled selector?** To justify our design choices, we evaluated alternative strategies, including head-only tuning and mimicking LLM's scores on the candidates. We find that head-only updates provide insufficient expressive capacity, while matching the LLM's scores overwhelms the student's reserved logit space with excessive noise. Consequently, we fine-tune SLM via a discrete selection loss, with a comprehensive analysis provided in Appendix D.4.

## 6. Conclusion

In this work, we argue that the primary reasoning bottleneck in SLMs is not a lack of generation potential, but a misalignment in candidate ranking. Our investigation into *local sufficiency* reveals that SLMs often generate viable next tokens within their top-candidate set, yet superior options can be suppressed by a suboptimal internal ranking. Leveraging it, S2T reframes distillation from open-ended distribution matching to discriminative selection over a bounded support, providing a more efficient path to unlocking the model's inherent potential. Specifically, our distilled SLM improves the 1.5B model's Math Avg. by 24.1% relative gain, achieving performance comparable to 8-path self-consistency while operating in a single-trajectory, teacher-free manner. Beyond these empirical gains, S2T recasts the SLM as an active selector of its own proposals rather than a passive imitator, suggesting a path toward resource-efficient scaling.

## Acknowledgements

We acknowledge the EuroHPC Joint Undertaking for awarding this project access to the EuroHPC supercomputer LEONARDO, hosted by CINECA (Italy) and the LEONARDO consortium, through a EuroHPC AI and Data-Intensive Applications Access call EHPC-AI-2024A06-060.

## Impact Statement

This paper introduces a selection-based distillation framework to enhance the reasoning of small language models. Our work contributes to the development of resource-efficient AI systems, enabling capable reasoning on edge devices with significantly lower latency and energy costs compared to traditional collaborative and ensemble methods.

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

This appendix is organized to provide comprehensive methodological specifics and supplementary evidence. We begin by detailing the formal inference algorithm in Appendix A, followed by an extended discussion of related works to further contextualize S2T within the broader literature in Appendix B. Appendix C offers a thorough description of our evaluation setup. Technical depth regarding the distillation of the selector and the training protocols for the trigger head is provided in Appendices D and E, respectively. Finally, Appendix F presents additional experimental results to further substantiate the robustness of our framework.

# A. Inference Algorithm

Algorithm 1 formalizes the S2T inference procedure, detailing the transition from standard generation to targeted discriminative selection at critical reasoning steps.

---

**Algorithm 1** S2T: Select-to-Think Inference
---

1: **Input:** Context $x$, SLM $\theta_S$, LLM $\theta_L$, Candidate size $K$, Threshold $\tau$, Mode $\in \{\text{COLLABORATIVE}, \text{LOCAL}\}$, predefined bin values $\mathbf{v}$
2: **Output:** Generated sequence $y$.
3: Initialize state $s_0 \leftarrow x$, $t \leftarrow 1$.
4: **while** $y_{t-1} \neq \langle \text{EOS} \rangle$ **do**
5:     Compute logits $z_t \leftarrow \text{Forward}(s_t; \theta_S)$.
6:     Determine trigger state $g_t$.
7:     **if** $g_t = 1$ **then**
8:         ▷ *Trigger activated: selection mode*
9:         Generate candidate set $\mathcal{C}_t \leftarrow \text{Sample-}K\,(z_t)$.
10:         **if** Mode $==$ COLLABORATIVE **then**
11:             **// LLM-Involved Scoring**
12:             **for** each $c \in \mathcal{C}_t$ **do**
13:                 Query Teacher: $\mathcal{S}(c; s_t) \leftarrow p_{\theta_L}(c \mid s_t)$.
14:             **end for**
15:             Select best token: $y_t \leftarrow \arg\max_{c \in \mathcal{C}_t} \mathcal{S}(c; s_t)$.
16:         **else**
17:             **// SLM-Internal Scoring (Distilled)**
18:             **for** each $c \in \mathcal{C}_t$ **do**
19:                 Form augmented state: $s_t^c \leftarrow (x, y_{<t} \oplus c)$.
20:                 Extract reserved logits: $q \leftarrow \text{Softmax}(z_{\theta_S}(s_t^c)[\mathcal{R}])$.
21:                 Compute Score: $\mathcal{S}_{\text{local}}(c; s_t) \leftarrow \mathbf{v}^\top q$.
22:             **end for**
23:             Select best token: $y_t \leftarrow \arg\max_{c \in \mathcal{C}_t} \mathcal{S}_{\text{local}}(c; s_t)$.
24:         **end if**
25:     **else**
26:         ▷ *Standard decoding*
27:         $y_t \leftarrow \text{Greedy}(z_t)$ or $\text{Sample}(z_t)$.
28:     **end if**
29:     Update state: $s_{t+1} \leftarrow (s_t, y_t)$, $t \leftarrow t + 1$.
30: **end while**
31: **Return** sequence $y$.

---

# B. Extended Related Work

## B.1. Inference-time verification and reranking

A dominant paradigm for enhancing inference accuracy involves sampling multiple candidates (e.g., Best-of-$N$) and reranking them using external signals, such as outcome verifiers (Cobbe et al., 2021), process reward models (PRMs) (Lightman et al., 2024), or LLM-as-a-Judge (Zheng et al., 2023). While effective, these methods typically incur high latency due to

extensive sampling and heavy scoring loops. Self-certainty (Kang et al., 2025) leverages the intrinsic probability distribution of LM outputs to estimate response quality without external reward models. However, as documented in the original study, self-certainty-driven Best-of-N selection is less effective than self-consistency on mathematical tasks with the same sampling budget ($N$). Consequently, we adopt self-consistency as one representative baseline.

Unlike traditional verifiers that rerank complete solutions, S2T distills teacher logic into a token-level selector, enabling real-time trajectory correction within a single forward pass, bypassing the computational overhead of generating and scoring multiple sequences.

**Speculative decoding**  Speculative decoding accelerates LLM inference by using a smaller draft model to propose candidate tokens or continuations that are then verified by the target LLM (Leviathan et al., 2023; Sun et al., 2023; Miao et al., 2023). Its primary goal is computational acceleration: the target LLM remains the generator whose behavior should be preserved, while the draft model serves as a proposal mechanism to reduce the cost of decoding from that target model.

Collaborative S2T has a superficial resemblance to single-step speculative verification, since an SLM proposes local candidates and an LLM evaluates the next-token decision. However, the role of the LLM and the purpose of the draft model are fundamentally different. In speculative decoding, the draft model is useful to the extent that its proposals can be accepted to speed up generation from the target LLM; the target LLM still defines the desired output behavior. In S2T, the SLM is the model whose reasoning ability we seek to improve, and the LLM is used only as a local evaluator over the SLM's own bounded candidate set at triggered positions.

This distinction changes the intervention from target-model generation to SLM-centered selection. Rather than asking the LLM to continue the reasoning trajectory from its full vocabulary, S2T asks whether the SLM's local support already contains a teacher-preferred decision. When this holds, the useful teacher signal can be reduced to ranking a small set of SLM-proposed candidates, revealing local sufficiency: the SLM often has the right option available locally but fails to select it as top-1. The collaborative S2T setting therefore serves to test and exploit this bounded-support sufficiency, while S2T-LOCAL further distills the resulting selection behavior into the SLM so that no LLM verification is required at test time.

### B.2. Contrastive decoding

A parallel line of research steers generation by modifying the decoding distribution itself, rather than employing external search. Foundational contrastive decoding formulations (Li et al., 2023) reshape logits by subtracting an "amateur" (or counterfactual) signal from an "expert" one to suppress generic or hallucinated tokens. This paradigm has inspired various extensions, including context-aware variants to resolve evidence-parametric conflicts (Shi et al., 2024), and internal-contrast methods (e.g., *DoLa* (Chuang et al., 2024)) that utilize intra-model signals to boost factuality.

Unlike methods that perform heuristic logit surgery across the full vocabulary, S2T operates on selection under bounded support. By explicitly ranking a restricted set of student-proposed candidates, we shift from passive distribution shaping to active path recovery.

## C. Evaluation Details

### C.1. Benchmark

**Mathematical reasoning benchmarks.**

- GSM8K (Cobbe et al., 2021): A dataset of 8,500 linguistically diverse grade school math word problems requiring 2-8 step reasoning, serving as a standard benchmark for evaluating basic mathematical reasoning capabilities.
- MATH-500 (Lightman et al., 2024): A subset of 500 challenging competition-level mathematics problems sampled from the MATH dataset, covering seven mathematical domains (algebra, geometry, number theory, etc.) ranging from AMC 8 to IMO difficulty levels.
- AIME (Zhang & Math-AI, 2025): American Invitational Mathematics Examination problems, featuring high-difficulty competition mathematics that requires advanced problem-solving skills and multi-step reasoning.
- OlympiadBench (He et al., 2024): A bilingual (English and Chinese) competition-level benchmark containing challenging problems from mathematical and scientific olympiads, testing advanced reasoning and problem-solving abilities.

**Comprehensive reasoning benchmarks.** MMLU-Pro (Wang et al., 2024): An enhanced version of MMLU (Massive Multitask Language Understanding) with more challenging questions across 14 domains including mathematics, physics,

computer science, and humanities, designed to reduce sensitivity to prompt variations and better assess reasoning capabilities.

**Code generation benchmark.** HumanEval (Chen et al., 2021): A benchmark of 164 hand-written programming problems for evaluating functional correctness of code synthesis, each consisting of a function signature, docstring, body, and unit tests.

We evaluate our models on a diverse set of reasoning benchmarks using a standardized zero-shot Chain-of-Thought (CoT) protocol. For mathematical datasets, we follow the evaluation protocol of Qwen2.5-Math (Yang et al., 2024). We extract the final answer (typically enclosed in boxed format) and compute Exact Match (EM) accuracy after standardizing formats using their official normalization scripts.

For HumanEval, we employ the pass@1 metric using the official evaluation framework from OpenAI's human-eval repository[1]. We format each problem prompt using the model's chat template with a system message instructing code-only output without markdown formatting or explanations. Code completions are generated with greedy decoding (temperature=0) and a maximum of 512 new tokens. We extract Python code from model outputs using regex patterns to remove markdown code blocks and truncate at subsequent function/class definitions. Functional correctness is verified by executing the generated code against provided unit tests in an isolated environment with a 3-second timeout per test case.

## C.2. Prompt

---
**Prompt for Mathematical benchmark**

```
Please reason step by step, and put your final answer within \boxed\{\}.
```
---

---
**Prompt for MMLU-Pro benchmark**

```
You are a helpful assistant. You are solving high-level academic problems. {question
    } Please reason step by step to find the correct answer. At the end of your
    response, output the final answer in the format: 'The answer is ({answer})'.
```
---

---
**Prompt for HumanEval benchmark**

```
You write Python code. Output only Python code. No markdown, no explanation.
```
---

## C.3. Implementation notes for representative baselines

We provide additional implementation notes for representative baselines in Table 1.

DISTILQWEN2.5 is an off-the-shelf industrial distilled model rather than a model retrained by us under the same data and optimization budget. It is trained via a two-stage knowledge distillation framework, initiating with a black-box phase where a multi-agent pipeline synthesizes, rigorously refines, and validates high-quality instruction data from proprietary models for supervised fine-tuning. Subsequently, an efficient white-box distillation phase mitigates industrial memory constraints and vocabulary mismatches by aligning tokens and minimizing predictive divergence exclusively over offline pre-computed top-K logits. We therefore include it as a practical reference for strong publicly available distilled performance, rather than as a controlled ablation.

TSD-KD is a student-centric on-policy distillation method in which the student first generates its own reasoning trajectory and the teacher then provides indirect distillation, selective direct distillation, and entropy regularization. In our reproduction, we use the same student–teacher pair, the same training dataset, and the same overall training volume as S2T-LOCAL, while following the optimization protocol of the published TSD-KD method. Thus, this comparison controls for teacher access and training scale, and mainly contrasts the learning target: TSD-KD distills toward the teacher's full-vocabulary behavior,

---
[1]https://github.com/openai/human-eval

whereas S2T-LOCAL reformulates supervision as bounded-support candidate selection.

LLM-TAKEOVER is a non-distillation collaborative ablation designed to isolate the intervention primitive. It uses the same trigger schedule and intervention budget as S2T, but replaces candidate selection with direct LLM token injection at triggered positions. This matched-budget comparison tests whether restricting the LLM to select among SLM proposals is sufficient relative to allowing the LLM to generate the next token directly.

## D. Implementation on Distilling Selector

Unless otherwise specified, all subsequent analyses default to evaluating Qwen2.5-1.5B-Instruct on the MATH500 dataset.

### D.1. Applicability of reserved tokens

S2T-LOCAL implements its ranking interface by repurposing a small number of reserved token IDs as scoring bins. This is a convenient implementation choice because many recent open-weight LLM families expose reusable reserved or padded vocabulary slots, often introduced for special-token expansion or hardware-friendly embedding dimensions. In our implementation, we can obtain the required reserved bins from commonly used families such as Qwen2.5, Llama 3.1/3.2, Phi-3, and Pythia without resizing the vocabulary or modifying the model architecture.

A small number of model families may not expose an obvious reusable bank of reserved IDs, e.g., Llama 2. In such cases, the same generation-selection decoupling can be implemented with a lightweight auxiliary ranking head parallel to the LM head. This fallback produces the same ranking scores and is trained with the same selection objectives, while adding negligible overhead compared with the base SLM.

### D.2. Training preparation

**Data collection.** To facilitate the internalization of the teacher's discriminatory capability, we construct a preference dataset derived exclusively from the training split of the MATH dataset. We perform data collection by rolling out the SLM. We align the SLM's internal ranking preferences with the teacher's probability landscape using a dataset of 2k instances, collected by targeting the top $\tau = 10\%$ of steps with the highest KL divergence; for each step, we sample $K = 16$ candidates annotated with the LLM's conditional probabilities.

**Hyperparameter configuration.** We train S2T-LOCAL with LoRA adapters applied to the attention and MLP projection modules, including `q_proj`, `k_proj`, `v_proj`, `o_proj`, `gate_proj`, `up_proj`, and `down_proj`. The selector predicts a categorical distribution over reserved scoring bins, which is converted into a scalar preference score using the fixed bin vector $v$. We use a score temperature of $T = 0.2$ when computing the ranking loss; this temperature is used for selector training and is distinct from decoding sampling temperature. The final training objective is

$$\mathcal{L} = \mathcal{L}_{\text{rank}} + \lambda_{\text{reg}}\mathcal{L}_{\text{reg}},$$

where $\mathcal{L}_{\text{reg}}$ is a prefix-level KL regularizer against the frozen base SLM. Unless otherwise specified, we set $\lambda_{\text{reg}} = 30$. For teacher denoising, we use margin filtering with mode drop and minimum margin $\delta = 0.08$. We configure the adapters with a rank of $r = 16$, a scaling factor $\alpha = 32$, and a dropout rate of $0.05$.

Crucially, to enable the model to learn the specific semantics of the reserved scoring bins ($\mathcal{R}$), we additionally unfreeze the specific lm$_{\text{head}}$ embeddings for the reserved tokens $\mathcal{R}$, with $|\mathcal{R}| = 16$. We utilize a contiguous block of 16 reserved tokens starting from ID 151920 - 151935. This ensures that repurposing their logits does not interfere with the model's linguistic capabilities. This configuration yields highly efficient adaptation: 35.7M and 74.3M trainable parameters for the 0.5B and 1.5B models, respectively ($< 6\%$ of total capacity). Unless otherwise stated, Section D reports the Qwen2.5-1.5B setting, for which we use $|\mathcal{R}| = 16$. For Qwen2.5-0.5B, the main experiments use $|\mathcal{R}| = 12$, as specified in Section 5.1. The model is optimized using AdamW with a global batch size of 64. We employ a cosine decay learning rate schedule starting from $10^{-4}$, preceded by a 3% linear warmup phase.

### D.3. Objective function

The model is trained to minimize a composite objective comprising the ranking loss, a stability regularizer, and an auxiliary margin loss. While Eq. (7) (Softmax Cross-Entropy) optimizes global alignment, we find that enforcing a strict separation

*Table A1. Hyperparameter sweep on Qwen2.5-1.5B-Instruct evaluated on MATH500.* FT Greedy denotes the greedy-decoding accuracy of the fine-tuned SLM, used as a proxy for retained base-model behavior.

| $|\mathcal{R}|$ | $\lambda_{\text{reg}}$ | S2T-LOCAL (%) | FT Greedy (%) | Agree@1 (%) |
|---|---|---|---|---|
| 8 | 30 | 61.4 | 50.6 | 48.5 |
| 12 | 30 | 69.8 | 55.1 | 66.9 |
| 16 | 5 | 50.4 | 32.5 | 70.4 |
| 16 | 30 | 67.5 | 49.6 | 67.1 |
| 16 | 70 | 58.4 | 49.6 | 63.2 |

between the best candidate and its "hard negatives" improves discrimination. We introduce an adaptive pairwise margin loss:

$$\mathcal{L}_{\text{margin}} = \text{ReLU}\Big(m_{\text{eff}} - \big(s(c^{\star}) - s(c^{\text{neg}})\big)\Big) \tag{13}$$

where $c^{\text{neg}}$ is the highest-scoring incorrect candidate (hard negative). Notably, we scale the margin dynamically based on the statistical dispersion of the candidate scores to avoid over-penalizing flat distributions. The effective margin is defined as:

$$m_{\text{eff}} = \frac{\sigma_t}{T} \tag{14}$$

where $\sigma_t$ is the standard deviation of scores in the current batch, and $T = 0.2$ is the selection-loss temperature controlling the sharpness of the candidate score distribution. The final loss is computed as $\mathcal{L} = \mathcal{L}_{\text{sel}} + \mathcal{L}_{\text{margin}} + \beta\mathcal{L}_{\text{KL}}$.

Empirically, the KL divergence term contributes a negligible fraction to the total loss, as the updated policy remains in close proximity to the reference model. To ensure effective regularization, we employ a substantial scaling coefficient $\beta$, typically parameterized within the range of 10-100.

### D.4. Ablation analysis

The training of the S2T selector was not a straightforward optimization but a multi-phase refinement process. We categorize the evolution into three distinct phases, each addressing a specific bottleneck in small-model distillation.

**Attempt I: initial failure modes.** Our initial attempts focused on a naive pointwise classification objective (predicting candidate correctness) using standard linear head-tuning. This approach failed due to two primary reasons: 1) Class imbalance: Because correct candidates are sparse in the student's proposals, the model optimized for the majority class (incorrect), resulting in a stagnant Agree@1 of $\approx 25\%$. 2) Vocabulary interference: The standard vocabulary embeddings lacked sufficient capacity to simultaneously support generation and discrimination, causing logit drift and linguistic instability.

**Attempt II: from distribution mimicry to decisive selection.** To overcome Phase I, we transitioned to fine-tuning SLM via LoRA, attempting to match the teacher's scores of candidates (i.e., learning the full ranking distribution via ListMLE). However, we observed that SLMs lack the capacity to internalize the teacher's full knowledge. Mimicking the teacher's "uncertainty" (soft-labels) resulted in a flat logit distribution, where the student could not confidently distinguish the top candidate from the runners-up. For capacity-constrained models, discrete selection (Top-1 Cross-Entropy) is superior. By focusing exclusively on the teacher's preferred choice, the student develops the sharp decision boundaries necessary for effective filtering.

**Attempt III: denoising and structural decoupling.** The final breakthrough involved stabilizing the training via teacher margin filtering: We discovered that not all teacher signals are beneficial. When the teacher exhibits low confidence (indicated by a narrow margin between Top-1 and Top-2 scores), its labels serve as noise rather than signal. By enforcing a margin threshold ($\delta = 0.08$), we filter for decisive teacher states, ensuring the student learns from high-quality supervision and significantly boosting Agree@1.

### D.5. Hyperparameter Sweep

We conduct a representative hyperparameter sweep to study the training sensitivity of S2T-LOCAL. The sweep focuses on two hyperparameters that significantly affect selector learning: the number of reserved scoring bins $|\mathcal{R}|$ and the stability regularizer $\lambda_{\text{reg}}$. The number of bins controls the resolution of the predicted scalar preference score, while $\lambda_{\text{reg}}$ weights the prefix-level KL penalty against the frozen base SLM and helps preserve the model's native behavior after fine-tuning.

*Table A2. Lightweight reserved-bin sweep across model sizes and architectures.* Other hyperparameters follow the default recipe.

| Model | $|\mathcal{R}|$ | S2T-LOCAL (%) | FT Greedy (%) | Agree@1 (%) |
|---|---|---|---|---|
| Qwen2.5, 1.5B | 12 | 69.8 | 55.1 | 66.9 |
| | 16 | 67.5 | 49.6 | 67.1 |
| Qwen2.5, 0.5B | 12 | 46.8 | 31.0 | 38.1 |
| | 16 | 43.4 | 27.6 | 35.6 |
| Llama3.2, 1B | 12 | 33.6 | 30.0 | 37.8 |
| | 16 | 34.6 | 29.2 | 41.9 |

Table A1 reports the main sensitivity sweep on Qwen2.5-1.5B-Instruct evaluated on MATH500. We report S2T-LOCAL accuracy, the greedy-decoding accuracy of the fine-tuned SLM (FT Greedy), and Agree@1 between the distilled selector and the teacher selector. Two trends are visible. First, $\lambda_{\mathrm{reg}}$ mainly controls stability: a too-small value, e.g., $\lambda_{\mathrm{reg}} = 5$, substantially degrades FT Greedy accuracy, while a too-large value, e.g., $\lambda_{\mathrm{reg}} = 70$, over-constrains selector learning. Second, $|\mathcal{R}| = 8$ under-resolves the ranking signal, whereas $|\mathcal{R}| = 12$ and $|\mathcal{R}| = 16$ both provide strong performance. Although $|\mathcal{R}| = 12$ gives the best MATH500 accuracy in this single sweep, we use $|\mathcal{R}| = 16$ and $\lambda_{\mathrm{reg}} = 30$ as the default configuration based on broader validation, since it provides a finer score resolution while maintaining comparable teacher alignment and retained base behavior.

We further examine whether the Qwen-derived recipe transfers across model sizes and architectures. For the non-Qwen setting, we use the Llama-3.x family, with Llama-3.2-1B-Instruct as the SLM and Llama-3.1-8B-Instruct as the LLM teacher. Most training hyperparameters, including the bin temperature, LoRA setup, and stability regularizer, transfer well without retuning. The main parameter that benefits from a lightweight sweep is the reserved-bin count $|\mathcal{R}|$, which slightly varies with model architecture and capacity. As shown in Table A2, both $|\mathcal{R}| = 12$ and $|\mathcal{R}| = 16$ are viable across Qwen and Llama settings, with the preferred choice depending mildly on the SLM itself. This supports using the Qwen-derived configuration as a strong default starting point, while treating $|\mathcal{R}|$ as the primary lightweight tuning knob for new model families.

# E. Divergence Point

Unless otherwise specified, all subsequent analyses default to evaluating Qwen2.5-1.5B-Instruct on the MATH500 dataset.

### E.1. Definition and calibration

A divergence point is a decoding step whose trigger score exceeds a calibrated threshold, e.g., the token-level KL divergence between LLM and SLM. The score is computed at every decoding step, and the threshold is set post-hoc on a small calibration set to match the target intervention budget. For deployment, S2T-LOCAL avoids runtime LLM dependence by using a lightweight MLP over the SLM hidden states to directly predict divergence scores.

### E.2. Trigger head

To enable learned triggering mechanisms that avoid the computational overhead of always computing KL divergence between SLM and LLM distributions, we train a lightweight trigger router that predicts critical decoding steps directly from the SLM's hidden states. The router consists of a criticality prediction head implemented as a two-layer MLP with ReLU activation and dropout (rate 0.1), mapping the SLM's hidden dimension to a scalar logit. Specifically, the architecture comprises: $\mathrm{Linear}(H \to 256) \to \mathrm{ReLU} \to \mathrm{Dropout}(0.1) \to \mathrm{Linear}(256 \to 1)$, where $H$ denotes the hidden dimension.

**Training data collection.** We collect training samples by performing SLM rollouts with KL-based triggering as the oracle signal. At each decoding step, we compute the KL divergence $D_{\mathrm{KL}}(p_{\theta_L} \| p_{\theta_S})$, and label the step as positive ($g_t = 1$) if $D_{\mathrm{KL}} > $ Top-$\tau$ Threshold and negative ($g_t = 0$) otherwise. For each step, we store the SLM's final-layer hidden state (dimension 1536 for Qwen2.5-1.5B-Instruct) along with its binary trigger label. The collected dataset exhibits significant class imbalance, with approximately 20% positive samples.

**Training configuration.** To address class imbalance, we employ Binary Cross-Entropy loss with positive class weighting: pos_weight $= n_{\mathrm{neg}}/n_{\mathrm{pos}} = 4.0$. The router is trained for 50 epochs using AdamW optimizer. We split the collected samples into 90% training and 10% validation sets with a fixed random seed (42) to ensure reproducibility. The best checkpoint is

*Table A3. Accuracy comparison under varying trigger threshold ($\tau$).* As the candidate budget $k$ scales, S2T consistently achieves performance parity with, and often surpasses the generative *Takeover* baseline across all benchmarks. **Bold** indicates the best performance within each $\tau$ block.

| Method | GSM8k | MATH | OlympiadBench | MMLU | Method | GSM8k | MATH | OlympiadBench | MMLU |
|---|---|---|---|---|---|---|---|---|---|
| **Budget $\tau = 1\%$** | | | | | **Budget $\tau = 2\%$** | | | | |
| Takeover | 94.9 | 80.0 | **38.1** | 54.0 | Takeover | 94.4 | 78.9 | 32.3 | **68.3** |
| S2T (2) | 91.1 | 71.1 | 33.2 | 44.4 | S2T (2) | 91.1 | 68.6 | 25.8 | 42.9 |
| S2T (4) | 95.6 | 79.8 | 31.6 | 49.2 | S2T (4) | **95.6** | **82.2** | 30.7 | 60.3 |
| S2T (8) | 95.6 | **81.1** | 37.5 | 55.6 | S2T (8) | 94.8 | 79.1 | **34.0** | 65.1 |
| S2T (16) | **95.7** | 80.0 | **38.1** | **58.7** | S2T (16) | 93.6 | 81.1 | 32.3 | 61.9 |
| | | | | | | | | | |
| **Budget $\tau = 5\%$** | | | | | **Budget $\tau = 10\%$** | | | | |
| Takeover | 94.0 | 78.3 | 40.3 | **71.0** | Takeover | **96.7** | 80.0 | 38.1 | **74.6** |
| S2T (2) | **96.7** | 68.9 | 27.4 | 55.6 | S2T (2) | 94.4 | 71.7 | 35.5 | 55.5 |
| S2T (4) | 96.3 | **80.0** | 41.9 | 66.7 | S2T (4) | 95.4 | 81.0 | **41.5** | 66.7 |
| S2T (8) | 94.4 | **80.0** | 40.3 | 68.3 | S2T (8) | 95.9 | **83.1** | 37.5 | 73.4 |
| S2T (16) | 95.6 | 75.6 | **42.0** | 70.1 | S2T (16) | 96.6 | 81.1 | 38.7 | 69.8 |
| | | | | | | | | | |
| **Budget $\tau = 20\%$** | | | | | | | | | |
| Takeover | 95.7 | 80.2 | 38.7 | 73.1 | | | | | |
| S2T (2) | 94.3 | 72.7 | 36.2 | 57.1 | | | | | |
| S2T (4) | **96.8** | 83.3 | 38.9 | 61.9 | | | | | |
| S2T (8) | 94.9 | **84.4** | **39.1** | 66.7 | | | | | |
| S2T (16) | **96.8** | **84.4** | **39.1** | 73.0 | | | | | |

selected based on validation accuracy, achieving *87.78% validation accuracy at epoch 36*. During inference, we apply a sigmoid activation to the criticality logit and threshold at 0.7 (configurable) to make binary triggering decisions, eliminating the need to compute expensive cross-model KL divergence at every step.

**Dynamic thresholding.** The 10% KL training split is used to curate "hard" examples for training the trigger head and is decoupled from the trigger budget used at inference time. To realize any target budget $\tau$ (e.g., 1% or 20%) at test time, we use a simple post-hoc percentile calibration: on a small held-out calibration set, we compute the trigger scores and set the threshold to the $(1 - \tau)$-th percentile. This threshold is then applied during inference, reliably approximating the target budget despite natural distribution shifts.

# F. Additional Experimental Results

For the remainder of our analysis, unless stated otherwise, the default experimental setup utilizes Qwen2.5-1.5B-Instruct on the MATH500 benchmark.

## F.1. Extensive analysis on the S2T performance compared to LLM Takeover

The results in Table A3 demonstrate that S2T performs competitively with the generative LLM-takeover baseline across all budgets, with performance parity or slight advantages depending on the candidate size k. This trend is robust across the entire spectrum of intervention budgets ($\tau$), suggesting that selection is a highly effective alternative to full-trajectory decoding even under stringent resource constraints.

Furthermore, we observe a clear scaling relationship between performance and the candidate set size ($K$). While a narrow candidate set often bottlenecks the teacher's corrective capacity, increasing $K$ consistently elevates accuracy by providing the selector with a more diverse range of alternatives to evaluate. This gain is particularly pronounced in complex reasoning tasks, where a larger K allows the teacher to identify correct tokens that the student ranked too low (e.g., beyond top-4) under greedy decoding. Across all budgets, S2T exhibits a convergence toward or beyond the generative ceiling as $K$ increases, establishing it as a scalable and computationally efficient framework for teacher-guided reasoning.

*Table A4. Effect of dynamic candidate sizing.* Dynamic-$K$ retains tokens whose probability is at least $1\%$ or $5\%$ of the top-token probability, whereas fixed $K = 8$ preserves substantially higher teacher-token coverage and achieves the best downstream accuracy across trigger rates.

| Trigger budget | Candidate strategy | Avg. $K$ | Hit@$K$ (%) | Acc. (%) |
|---|---|---|---|---|
| 1% | Dynamic (1%) | 3.1 | 76.0 | 75.0 |
| | Dynamic (5%) | 3.6 | 80.2 | 73.6 |
| | Fix. (8) | 8.0 | 96.3 | 81.1 |
| 2% | Dynamic (1%) | 3.1 | 81.4 | 77.5 |
| | Dynamic (5%) | 3.6 | 85.3 | 77.9 |
| | Fix. (8) | 8.0 | 97.0 | 79.1 |
| 10% | Dynamic (1%) | 2.5 | 94.1 | 77.8 |
| | Dynamic (5%) | 3.1 | 95.7 | 74.9 |
| | Fix. (8) | 8.0 | 99.5 | 83.1 |

*Table A5. Additional evaluation on non-math domains using the same checkpoints trained only on MATH.* TruthfulQA and ARC-Challenge report accuracy (%), while MT-Bench reports the judge score.

| Backbone | Benchmark | SLM Greedy | S2T-LOCAL | S2T |
|---|---|---|---|---|
| Qwen2.5-0.5B-Instruct | TruthfulQA | 18.2 | 34.2 | 54.3 |
| | ARC-C | 29.9 | 42.5 | 77.1 |
| | MT-Bench | 3.4 | 3.6 | 5.6 |
| Qwen2.5-1.5B-Instruct | TruthfulQA | 46.7 | 54.6 | 68.6 |
| | ARC-C | 61.0 | 73.9 | 88.6 |
| | MT-Bench | 4.8 | 5.1 | 5.6 |
| Llama3.2-1B-Instruct | TruthfulQA | 35.1 | 39.3 | 45.7 |
| | ARC-C | 45.5 | 49.2 | 58.6 |
| | MT-Bench | 4.3 | 4.2 | 4.8 |

**Effect of dynamic candidate sizing** We further study whether the candidate budget can be selected dynamically instead of using a fixed $K$. Specifically, we consider threshold-based dynamic candidate sizing, where a token is retained if its probability is at least $1\%$ or $5\%$ of the top-token probability at a triggered position. Although this strategy reduces the average candidate size to roughly $K \in [2.0, 3.6]$, it is less reliable at critical decision points, where the SLM distribution is often miscalibrated and the teacher-preferred token may lie outside the high-probability prefix. As shown in Table A4, dynamic candidate sizing consistently yields lower teacher-token coverage and lower downstream accuracy than the fixed $K = 8$ strategy across trigger rates. Together with the fixed-$K$ sweep in Table A3, these results indicate that $K = 8$ offers a robust trade-off between preserving local support and avoiding unnecessary candidate expansion.

### F.2. Additional evaluation on non-math domains

We further examine whether S2T-LOCAL, trained only on MATH, transfers to broader non-math domains. To isolate cross-domain transfer rather than task-specific retraining, we evaluate the same checkpoints on TruthfulQA (Lin et al., 2022) for factuality, ARC-Challenge (Clark et al., 2018) for challenging science question answering, and MT-Bench (Zheng et al., 2023) for open-ended instruction following. As shown in Table A5, S2T-LOCAL consistently improves over greedy decoding on TruthfulQA and ARC-C across the three instruction-tuned SLM backbones, while maintaining comparable or slightly improved open-ended generation quality on MT-Bench. Averaged over these backbones, S2T-LOCAL yields relative gains of $38.9\%$ on TruthfulQA, $23.8\%$ on ARC-C, and $3.3\%$ on MT-Bench over SLM greedy decoding. These results suggest that preserving local candidate diversity at trigger points does not introduce systematic regressions in broader factuality or generation quality, while the learned selector remains effective beyond the math-only training domain.

### F.3. Run-to-run stability

We further report the run-to-run variation of the main Greedy and S2T-LOCAL results across independent runs with different random seeds. Table A6 shows the mean and standard deviation on the six benchmarks used in Table 1. Across most settings, the standard deviations are small relative to the absolute improvements of S2T-LOCAL over Greedy, indicating that the observed gains are robust to random-seed variation rather than being driven by isolated runs.

*Table A6. Run-to-run stability of Greedy and* S2T-LOCAL *across independent random seeds.* We report mean $\pm$ standard deviation.

| Model | Method | GSM8K | MATH500 | OlympiadBench | AIME25 | HumanEval | MMLU-Pro |
|---|---|---|---|---|---|---|---|
| 0.5B | Greedy | 44.5±1.4 | 30.4±1.6 | 5.8±0.6 | 0.0±0.0 | 18.9±1.6 | 11.3±0.3 |
| | S2T-LOCAL | 61.6±1.2 | 46.8±0.8 | 8.5±1.4 | 0.0±0.0 | 26.7±2.2 | 18.7±1.2 |
| 1.5B | Greedy | 72.1±1.2 | 54.4±2.2 | 20.1±0.7 | 1.1±1.5 | 42.7±1.8 | 22.9±1.4 |
| | S2T-LOCAL | 86.6±1.0 | 67.5±1.5 | 27.0±2.4 | 2.2±0.5 | 56.9±2.0 | 33.6±1.2 |

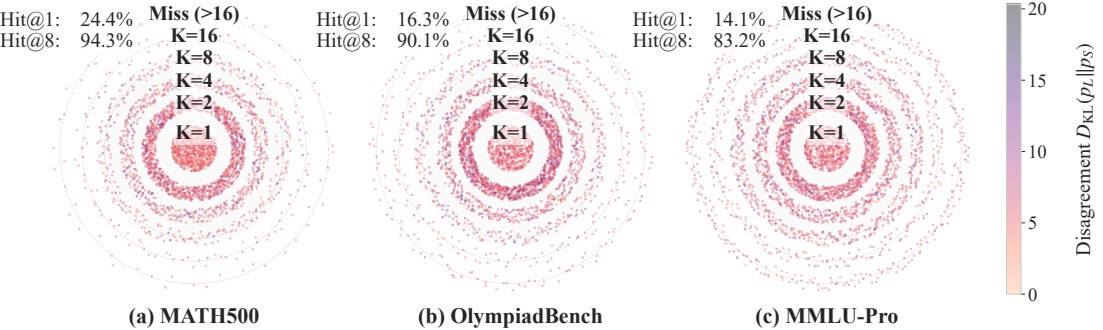

*Figure A1. Validation of local sufficiency on the 0.5B SLM.* Radial visualization of minimal $K$ required to match LLM targets across (a) MATH500, (b) OlympiadBench, and (c) MMLU-Pro. Despite the smaller model scale, the dense central clustering persists: a compact candidate set with $K = 8$ contains the LLM-preferred token in over 80% of cases. The consistent patterns across both reasoning and general knowledge (MMLU-Pro) confirm that LLM-grade tokens are inherently present in the SLM's distribution but typically bypassed by greedy decoding.

## F.4. S2T-LOCAL training on Llama

To further test whether the selection behavior can be distilled into a non-Qwen student, we train the complete S2T-LOCAL pipeline on the Llama-3.x family, using Llama-3.2-1B-Instruct as the SLM and Llama-3.1-8B-Instruct as the LLM teacher. Consistent with the main Qwen setup, the S2T-LOCAL model is trained only on MATH data and performs inference without LLM calls at test time. As shown in Table A8, S2T-LOCAL improves over greedy decoding on all evaluated benchmarks, while collaborative S2T provides a strong upper bound. This confirms that the distilled local selector transfers beyond Qwen and can improve a fully teacher-free Llama student.

We also report the corresponding local-sufficiency and selector-alignment diagnostics. The large gap between Hit@1 and Hit@8 is preserved across datasets, indicating that local sufficiency remains valid for the Llama architecture. Meanwhile, the distilled selector achieves non-trivial Agree@1 with the teacher selector while maintaining reasonable FT Greedy accuracy, suggesting that the learned selector improves local decisions without simply overwriting the student's base reasoning behavior.

## F.5. Validity of local sufficiency

To further verify the robustness of the "local sufficiency" hypothesis, we extended our radial visualization analysis to a much smaller scale (Qwen2.5-0.5B-Instruct) and a distinct model family (Gemma-2-2B-IT, Gemma-2-27B-IT (Team et al., 2024)). As illustrated in Figure A1 and Figure A2, a striking consistency emerges: despite the significant disparity in model parameters and architectural designs (e.g., different tokenizers and training recipes), the dense central clustering remains invariant. In over 80% of instances across all tested benchmarks, the LLM-preferred tokens consistently reside within the SLM's top-8 candidate set.

As illustrated in Figure A3, we conduct a fine-grained analysis of the rank distribution of LLM-preferred tokens within the SLM's logit distribution. A critical observation is the dramatic surge in coverage when expanding the candidate set from $K = 1$ to $K = 8$. Specifically, in the most challenging scenarios where the SLM's greedy hit rate (Hit@1) falls to a mere 14.1%, the inclusion of just 7 additional candidates (up to $K = 8$) successfully recovers the LLM's target in 83.2% of cases. This 69.1% absolute gain in the "worst-case" scenario, coupled with the 96.3% peak coverage in other tasks, demonstrates that the bottleneck of SLMs is not a fundamental lack of knowledge, but a systematic selection failure during decoding. Furthermore, this pattern remains remarkably consistent across both the 0.5B and 1.5B scales, suggesting that

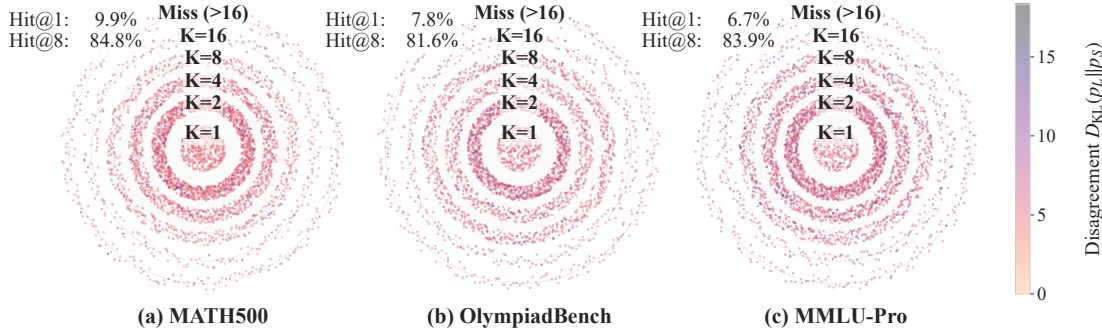

(a) MATH500      (b) OlympiadBench      (c) MMLU-Pro

*Figure A2. Validation of local sufficiency on Gemma model family with SLM being Gemma-2-2B-IT, LLM being Gemma-2-27B-IT.* The results mirror the findings on the Qwen-2.5 series: even under a different architecture and tokenizer, a compact candidate set with $K = 8$ still contains the target in over 80% of instances. This suggests that *Local Sufficiency* is a model-agnostic property of SLMs rather than an artifact of a specific model family.

*Table A7. Local sufficiency and collaborative* S2T *across non-Qwen model families.* Hit@8 measures whether the teacher-preferred token is contained in the SLM's top-8 candidate set at triggered positions.

| Model Pair | Data | Hit@1 (%) | Hit@8 (%) | SLM Greedy (%) | S2T (%) |
|---|---|---|---|---|---|
| Llama 3.x (1B → 8B) | MATH500 | 23.3 | 92.2 | 27.8 | 45.5 |
| | MMLU-Pro | 21.7 | 86.4 | 19.5 | 28.6 |
| Phi-3 (3.8B → 14B) | MATH500 | 31.4 | 97.5 | 40.9 | 54.6 |
| | MMLU-Pro | 26.3 | 93.5 | 27.1 | 41.3 |
| Gemma 2 (2B → 27B) | MATH500 | 9.9 | 84.8 | 31.2 | 40.4 |
| | MMLU-Pro | 6.7 | 83.9 | 14.1 | 23.2 |

local sufficiency is a recurring property of the tested SLM token distributions, rather than an artifact of a single model size or dataset.

We provide the rank distribution of selected tokens within the SLM's prediction space in Figure A4. The CDF plot reveals a critical insight: the correct tokens are not absent, but hidden within the immediate search scope. While the SLM often fails to assign the highest probability to the optimal token (low Rank-1 accuracy), coverage rises sharply within the top-3 ranks, often exceeding 80% and remaining above 70% even in the most challenging setting. It indicates that the model is suffering from decision ambiguity among a few strong candidates, rather than lacking the requisite knowledge.

Beyond the visualization-based analysis, we further evaluate local sufficiency across additional model families using three student–teacher pairs: Llama-3.2-1B-Instruct → Llama-3.1-8B-Instruct, Phi-3-3.8B → Phi-3-14B, and Gemma-2-2B-IT → Gemma-2-27B-IT. For all pairs, we use the default trigger setting based on the top 1% KL-divergence positions and evaluate on MATH500 and MMLU-Pro. Table A7 reports Hit@1, Hit@8, SLM greedy accuracy, and collaborative S2T accuracy. Across all three non-Qwen families, Hit@8 is substantially higher than Hit@1, confirming that the teacher-preferred token is often not the SLM's greedy choice but remains within its local top-8 candidate set. This provides tabular evidence consistent with the CDF and radial visualizations: local sufficiency is not specific to Qwen, but persists across different model families, tokenizers, and training recipes. Correspondingly, collaborative S2T consistently improves over SLM greedy decoding, indicating that exposing and selecting from the SLM's bounded local support is beneficial beyond the original Qwen setting.

### F.6. Performance of distilled selector

To provide a rigorous analysis of the selector's performance, we first define the evaluation metrics utilized in Table A9:

- Selection Alignment: includes Agree@1, which measures the frequency with which the selector's top choice matches the teacher's preferred candidate; Kendall's $\tau$ and Spearman $\rho$ quantify the rank correlation between the distilled scoring and the teacher's probability distribution.
- Confidence on selected tokens: compares the probability assigned to the selected token by the base SLM, the LLM (teacher), and the distilled selector (Score).

*Table A8.* S2T *evaluation on Llama-3.2-1B with Llama-3.1-8B as teacher.* S2T-LOCAL is trained only on MATH and uses no LLM calls at test time.

| Dataset | Performance Metrics | | | Detailed Analysis | | | |
|---|---|---|---|---|---|---|---|
| | SLM Greedy | S2T-LOCAL | S2T | Hit@1 | Hit@8 | FT Greedy | Agree@1 |
| GSM8K | 37.5 | 43.6 | 56.7 | 27.7 | 85.1 | 40.0 | 34.9 |
| MATH500 | 27.8 | 34.6 | 45.5 | 23.3 | 92.2 | 29.2 | 41.9 |
| OlympiadBench | 2.5 | 6.4 | 8.1 | 27.2 | 87.9 | 4.8 | 33.6 |
| MMLU-Pro | 16.5 | 20.3 | 28.6 | 21.7 | 86.4 | 17.5 | 29.6 |

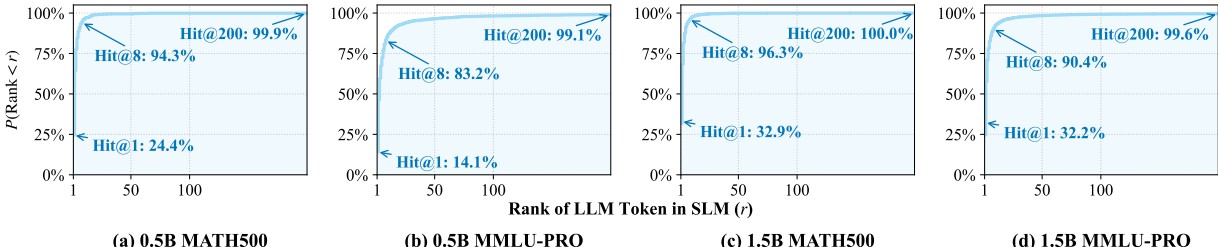

| (a) 0.5B MATH500 | (b) 0.5B MMLU-PRO | (c) 1.5B MATH500 | (d) 1.5B MMLU-PRO |

*Figure A3. Cumulative distribution of LLM top-1 token ranks in SLM vocabulary distribution.* We plot the CDF of the rank of LLM-preferred tokens within the SLM vocabulary distribution of 0.5B (left) and 1.5B (right) across MATH500 and MMLU-Pro. Despite a fragile greedy Hit@1 that drops to 14.1% in complex reasoning contexts, the local proposal space at $K = 8$ retains high fidelity, with coverage consistently reaching 83.2%–96.3%. This robust recovery across diverse scales and domains validates the *Local Sufficiency* hypothesis, indicating that the target knowledge is preserved in the SLM's local neighborhood even when its top-1 prediction fails.

*Table A9. Robustness of selector alignment.* The distilled selector achieves non-trivial fidelity with teacher preferences across diverse benchmarks and model scales. For the 1.5B model, it often assigns high scores to teacher-preferred tokens that the base SLM underestimates, while the 0.5B model still shows meaningful rank-correlation gains despite a larger capacity gap.

| Scale | Metric | | Benchmark | | | | | | Random |
|---|---|---|---|---|---|---|---|---|---|
| | | | GSM8k | MATH500 | OB | AIME25 | HumanEval | MMLU-Pro | |
| | **selection Alignment** | Agree@1 (↑) | 38.7% | 38.1% | 38.7% | 37.6% | 34.6% | 35.3% | 12.5% |
| | | Kendall's $\tau$ (↑) | 0.48 | 0.50 | 0.49 | 0.49 | 0.44 | 0.40 | 0.00 |
| | | Spearman $\rho$ (↑) | 0.60 | 0.61 | 0.60 | 0.61 | 0.54 | 0.49 | 0.00 |
| **0.5B** | **Confidence on selected tokens** | SLM probs | 43.9% | 46.4% | 45.7% | 45.0% | 22.6% | 41.3% | - |
| | | LLM probs | 31.7% | 31.6% | 33.7% | 32.2% | 30.4% | 25.4% | - |
| | | Score | 71.1% | 70.0% | 65.5% | 66.1% | 79.1% | 76.4% | - |
| | **Prediction confidence** | LLM entropy | 0.08 | 0.08 | 0.12 | 0.12 | 0.06 | 0.19 | - |
| | | score entropy | 1.32 | 1.27 | 1.20 | 1.22 | 1.82 | 1.35 | - |
| | **selection Alignment** | Agree@1 (↑) | 72.3% | 69.4% | 71.4% | 69.3% | 63.2% | 67.2% | 12.5% |
| | | Kendall's $\tau$ (↑) | 0.59 | 0.54 | 0.54 | 0.55 | 0.54 | 0.49 | 0.00 |
| | | Spearman $\rho$ (↑) | 0.67 | 0.63 | 0.65 | 0.67 | 0.62 | 0.61 | 0.00 |
| **1.5B** | **Confidence on selected tokens** | SLM probs | 26.2% | 28.5% | 28.4% | 31.2% | 23.3% | 25.2% | - |
| | | LLM probs | 66.3% | 69.9% | 67.0% | 62.2% | 38.5% | 45.4% | - |
| | | Score | 92.0% | 84.2% | 80.3% | 74.7% | 81.8% | 77.3% | - |
| | **Prediction confidence** | LLM entropy | 0.05 | 0.07 | 0.09 | 0.11 | 0.09 | 0.19 | - |
| | | Score entropy | 1.56 | 1.43 | 1.29 | 1.32 | 1.37 | 1.21 | - |

- Prediction confidence: measured via entropy, where lower values indicate a more peaked or certain distribution; this metric compares the teacher's certainty with the selector's scoring sharpness.

The results demonstrate that the distilled selector achieves substantial alignment fidelity with teacher preferences across all benchmarks, significantly outperforming the Random baseline (12.5%). While performance scales with model capacity, as the 1.5B model reaches over 70% Agree@1 accuracy, the 0.5B variant still maintains strong rank correlations ($\rho \approx 0.50$), suggesting that even extremely small models can internalize complex ranking hierarchies.

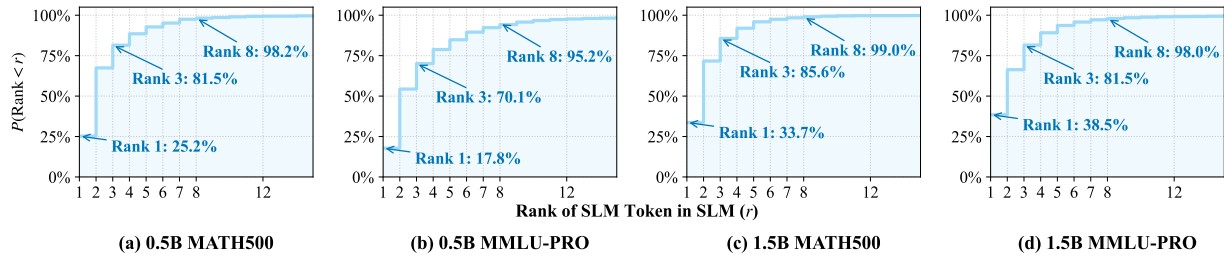

| (a) 0.5B MATH500 | (b) 0.5B MMLU-PRO | (c) 1.5B MATH500 | (d) 1.5B MMLU-PRO |

*Figure A4. CDF of the rank of selected tokens within the SLM prediction space.* Across model sizes and datasets, coverage rises sharply within the top few ranks, showing that the relevant tokens are usually not absent from the SLM distribution but are hidden among a small set of plausible candidates. The top-3 coverage often exceeds 80% and remains above 70% even in the most challenging setting, supporting the view that SLM failures frequently arise from local selection ambiguity rather than complete knowledge absence.

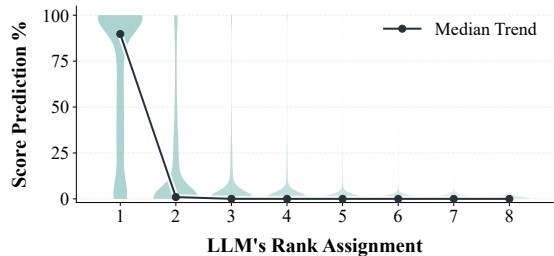

*Figure A5. Distribution of SLM-predicted scores across LLM-assigned ranks.* The violin plots show the density of scores, and the line tracks the median trend. The distilled selector acts as a decisive binary classifier, assigning high probability mass almost exclusively to the teacher's top choice while aggressively suppressing lower-ranked candidates.

Across diverse tasks ranging from linguistic reasoning (MMLU-Pro) to formal mathematics (AIME25), the selector exhibits robust stability, confirming that the SELECT TO THINK paradigm is effective regardless of the underlying student scale. A critical finding lies in the calibration recovery observed in the confidence metrics, particularly for the 1.5B model. The data reveals that the selector frequently identifies tokens where the base SLM probability is low ($\approx 25 - 30\%$) but the LLM probability is high ($\approx 60 - 70\%$). By elevating the Score probability to over $80 - 90\%$, the selector effectively "rescues" optimal candidates that were previously underestimated by the student's original distribution. This demonstrates that S2T does not merely mimic the student's existing biases, but successfully adopts the teacher's discriminative logic to overcome the student's inherent calibration failures.

To gain a deeper understanding of the selector's behavior, we visualize the distribution of predicted scores against the ground-truth ranks assigned by the teacher in Figure A5. The violin plot reveals a striking "winner-takes-all" characteristic: *High fidelity on top-1:* The distribution for Rank 1 is heavily skewed towards the top (median $\approx 0.9$), indicating that the selector consistently assigns high confidence to the teacher's preferred token. *Sharp discrimination:* We observe a precipitous drop in scores from Rank 1 to Rank 2, with the median score plunging to near zero. This suggests that the selector does not merely learn a soft correlation; instead, it establishes a sharp decision boundary, effectively suppressing sub-optimal candidates (Ranks 2-8) to eliminate ambiguity. This decisive separation explains why the S2T-LOCAL model achieves high performance even with a small candidate set: it acts as a high-precision filter that decisively isolates the optimal path.

**Analysis of selector disagreement**   We further analyze the disagreement between the distilled selector in S2T-LOCAL and the LLM teacher selector in S2T. Although the distilled selector does not perfectly match the teacher at every triggered step, not all disagreements are equally harmful: some correspond to stylistic alternatives or valid reordered reasoning steps. However, disagreement is strongly associated with downstream failure and therefore helps explain the remaining gap between S2T-LOCAL and S2T.

On the MATH500 and GSM8K validation sets, we partition trajectories into an *agree* group, where the distilled selector has zero disagreements with the teacher across all triggered steps, and a *disagree* group, where at least one triggered step disagrees. Across 900 sampled cases, the agree group achieves 96.1% final accuracy, whereas the disagree group drops to 35.3%. Moreover, the number of selector–teacher disagreement steps per trajectory is negatively correlated with final

*Table A10. Ablation of trigger metrics.* For each trigger metric and intervention threshold, S2T is compared against a matched *Takeover* baseline that uses the same trigger positions and budget but lets the LLM directly generate the next token. Across trigger settings, S2T remains broadly competitive with *Takeover*, showing that bounded selection over SLM candidates can provide similar benefit to generative LLM intervention when applied at the same positions.

| Trigger | Method | Trigger Threshold ($\tau$) | | | | |
|---|---|---|---|---|---|---|
| | | **1%** | **2%** | **5%** | **10%** | **20%** |
| **Random** | Takeover | 58.0 | 57.1 | 54.2 | 59.6 | 63.4 |
| | S2T | 64.0 | 61.2 | 58.3 | 61.7 | 63.0 |
| **SLM Entropy** | Takeover | 58.9 | 58.9 | 60.5 | 71.4 | 80.5 |
| | S2T | 56.5 | 54.5 | 65.1 | 76.2 | 78.5 |
| **KLD** | Takeover | 80.0 | 78.9 | 78.3 | 80.0 | 80.2 |
| | S2T | 81.1 | 79.1 | 80.0 | 83.1 | 84.4 |

*Table A11. Effect of trigger metrics on local sufficiency.* We compare trigger metrics under a fixed top-1% intervention budget. Hit@8 remains high across metrics and datasets, showing that local sufficiency is not tied to a specific trigger definition. However, downstream S2T accuracy is highest with KL divergence, indicating that targeted triggers better identify positions where local selection can correct student failures.

| Dataset | Trigger Metric | Hit@1 (%) | Hit@8 (%) | S2T Acc. (%) |
|---|---|---|---|---|
| | KL Divergence | 32.9 | 96.3 | 81.1 |
| MATH500 | SLM Entropy | 40.0 | 95.8 | 56.5 |
| | Random | 93.4 | 100.0 | 55.5 |
| | KL Divergence | 32.2 | 91.5 | 30.2 |
| MMLU-Pro | SLM Entropy | 39.6 | 91.4 | 22.2 |
| | Random | 89.2 | 99.0 | 23.8 |

correctness ($r = -0.46$, $p = 1.40 \times 10^{-44}$). These results show that selector disagreement is a meaningful diagnostic signal for residual errors in the distilled local selector.

Qualitative inspection suggests that disagreement cases are heterogeneous. Benign disagreements often involve stylistic or discourse-level alternatives, such as choosing "So" instead of "Now", or alternative but valid orderings of intermediate reasoning steps. In contrast, harmful disagreements typically occur at execution-critical decisions, including wrong arithmetic operators, incorrect numerical values, or branch choices that send the solution along an invalid reasoning path. This suggests that future improvements to S2T-LOCAL may benefit from emphasizing high-impact trigger positions where local selection errors are most likely to affect the final answer.

### F.7. Ablation on trigger mechanism

Autoregressive trajectories can shift under minor perturbations, so exact token-level intervention locations are not expected to remain fixed. We therefore evaluate whether local sufficiency remains robust when the trigger metric or trigger budget changes. Table A10 compares S2T with a matched *Takeover* baseline under the same trigger metric and intervention budget. Although different trigger metrics lead to different absolute accuracies, S2T remains broadly competitive with *Takeover* within each matched setting. This suggests that the main advantage of LLM intervention at triggered positions can often be recovered through bounded candidate selection, without requiring the LLM to generate the next token directly.

Table A11 further shows that Hit@8 remains high across trigger metrics, indicating that local sufficiency is not tied to a particular trigger heuristic, although selecting useful intervention points remains important for translating candidate coverage into final accuracy. We further vary the trigger budget $\tau$ for KLD and entropy triggers on MATH. As shown in Table A12, increasing the trigger budget consistently raises Hit@8 for both metrics, while KLD achieves stronger downstream accuracy under small budgets. This supports our use of KLD as the default trigger metric: it better identifies decision points where bounded-support selection is both available and useful.

### F.8. Extended efficiency analysis

The efficiency analysis in Table A13 underscores the practical viability of our framework, revealing that S2T-LOCAL successfully eliminates the prohibitive latency overhead typically associated with teacher-student collaboration. While

*Table A12. Robustness to trigger budgets on MATH.*

| Trigger Metric | Metric | $\tau = 1\%$ | $\tau = 5\%$ | $\tau = 20\%$ |
|---|---|---|---|---|
| KLD | S2T Acc. (%) | 81.1 | 80.0 | 84.4 |
| KLD | Hit@8 (%) | 96.3 | 98.8 | 99.8 |
| SLM Entropy | S2T Acc. (%) | 56.5 | 65.1 | 78.5 |
| SLM Entropy | Hit@8 (%) | 95.8 | 99.0 | 99.7 |

*Table A13. Efficiency analysis on average generated tokens and inference latency (seconds/problem).* S2T-LOCAL matches standard SLM inference speeds, delivering a ∼75% time reduction over the collaborative methods while maintaining generated token counts.

| Method | GSM8k | | MATH500 | | OlympiadBench | | AIME25 | |
|---|---|---|---|---|---|---|---|---|
| | Tokens | Time (s) | Tokens | Time (s) | Tokens | Time (s) | Tokens | Time (s) |
| *Baselines* | | | | | | | | |
| SLM (Greedy) | 329.5 | 6.1 | 493.8 | 9.3 | 827.5 | 15.5 | 1631.6 | 30.7 |
| LLM (Greedy) | 300.3 | 17.8 | 534.6 | 31.9 | 647.2 | 38.5 | 1229.3 | 73.7 |
| SpecReason | 300.1 | 11.8 | 469.5 | 23.2 | 730.3 | 54.7 | 1052.2 | 65.5 |
| Takeover | 297.6 | 23.1 | 518.1 | 40.5 | 938.8 | 74.7 | 1028.7 | 81.5 |
| *Ours* | | | | | | | | |
| S2T | 297.1 | 23.6 | 513.3 | 40.5 | 870.8 | 69.7 | 1079.8 | 85.8 |
| S2T-LOCAL | 300.1 | 6.6 | 512.4 | 10.2 | 903.3 | 18.0 | 1053.8 | 21.0 |
| *vs. Takeover* | +0.8% | -71.4% | -1.1% | -74.8% | -3.8% | -75.9% | +2.4% | -74.2% |

standard collaborative methods like *SpecReason* and *LLM-takeover* incur significant delays due to real-time teacher intervention, S2T-LOCAL achieves inference latencies nearly identical to that of standard SLM (Greedy) decoding. Specifically, it delivers a consistent ∼75% time reduction compared to the *LLM-takeover* baseline across all benchmarks, notably reducing AIME25 latency from 81.5s to just 21.0s. Crucially, this massive acceleration is achieved without a corresponding reduction in generated token counts, proving that the efficiency gains stem entirely from internalizing the selection logic into the local model rather than simply truncating the reasoning paths. By bypassing the need for expensive online teacher calls, S2T-LOCAL effectively provides a significant performance boost, maintaining the native inference speed of small-scale models while significantly elevating their reasoning accuracy.

Interestingly, as shown in Table A14, the time consumption remains remarkably stable across various trigger rates and $K$ values (averaging around 10.0s), with only a marginal overhead compared to the SLM greedy baseline (9.3s).

## F.9. Full trajectory example

To illustrate the selection dynamics and local sufficiency of S2T, we visualize a reasoning trajectory, where highlighted tokens indicate distributional divergence between the SLM and LLM:

---

**Math problem under S2T**

- **Blue**: The SLM and LLM agree on the Top-1 token despite differing confidence levels.

- **Green**: The teacher validates the SLM's Top-1 as the best option within the local pool (even if its global preference differs).

- **Red**: Critical intervention points where the LLM selects a token that is not the SLM's Top-1.

To solve the problem, we start by analyzing the polynomial $x^8 + 3x^4 - 4$. We can rewrite it as:

$$x^8 + 3x^4 - 4 = (x^4 + \boxed{4})(x^4 - 1).$$

Next, we factor $x^4 \boxed{+} 4$ and $x^4 - 1$ further $\boxed{.}$ Notice that:

$$x^4 + 4 = (x^2 + 2x + 2)(x^2 - 2x + 2),$$

and

$$x^4 - 1 = (x^2 + 1)(x^2 - 1) = (x^2 + 1)(x + 1)(x - 1).$$

---

*Table A14. Efficiency comparison* (time consumption in seconds) across different trigger rates $\tau$ and candidate sizes $K$. The time consumption remains remarkably stable and introduces only marginal overhead relative to the SLM greedy baseline.

| Method | Trigger Rate ($\tau$) | | | | |
|---|---|---|---|---|---|
| | **1%** | **2%** | **5%** | **10%** | **20%** |
| SLM greedy | | | 9.3 | | |
| S2T-LOCAL ($K = 2$) | 10.0 | 9.9 | 10.0 | 10.0 | 9.8 |
| S2T-LOCAL ($K = 4$) | 9.9 | 10.0 | 10.0 | 10.2 | 9.7 |
| S2T-LOCAL ($K = 8$) | 10.1 | 10.0 | 10.0 | 10.1 | 9.8 |
| S2T-LOCAL ($K = 16$) | 10.0 | 9.8 | 10.1 | 10.0 | 10.0 |

*Table A15. Impact of sampling parameters at trigger points.* Restrictive decoding (e.g., $T=0.6, p=0.7$) constrains the unique candidate count $|\mathcal{C}_t|$, resulting in low coverage and suboptimal performance. S2T utilizes diversity-oriented sampling ($T=1.0, p=1.0$) to ensure a full-size candidate pool ($|\mathcal{C}_t|=8$), yielding superior downstream accuracy.

| Method | Temp ($T$) | Top-$p$ | Avg. $|\mathcal{C}_t|$ | Hit@8 % | Acc. |
|---|---|---|---|---|---|
| Sampling | 0.6 | 0.70 | 1.5 | 37.6 | 57.6 |
| | | 0.95 | 2.7 | 66.2 | 66.0 |
| | 1.0 | 0.70 | 2.0 | 50.1 | 71.7 |
| | | 0.95 | 3.9 | 85.1 | 71.7 |
| S2T | 1.0 | 1.0 | 8.0 | 97.3 | 81.1 |

```
Thus, we can write:
```

$$x^8 + 3x^4 - 4 = (x^2 + 2x + 2)\,(x^{\boxed{2}} - 2x + 2)\,(x^{\boxed{2}} + 1)\,(x+1)\,(x-1).$$

```
The pol ynomials x² + 2 x + 2, x² − 2x + 2, x² + 1, x + 1, and x − 1 are all mon ic and have integer
coefficients. They cannot be factored further over the integers. Therefore, the pol ynomials
p₁(x), p₂(x), p₃(x), p₄(x), and p₅(x) are:
```

$$p_1(x) = x^2 + 2x + 2,$$

$$p_2(x) = x^2 - 2x + 2,$$

$$p_3(x) = x^2 + 1,$$

$$p_4(x) = x + 1,$$

$$p_5(x) = x - 1.$$

```
We need to compute p₁(1) + p₂(1) + p₃(1) + p₄(1) + p₅(1). Evaluating each polynomial at x = 1, we get:
```

$$p_1(1) = 1^2 + 2 \cdot 1 + 2 = 1 + 2 + 2 = 5,$$

$$p_2(1) = 1^2 - 2 \cdot 1 + 2 = 1 - 2 + 2 = 1,$$

$$p_3(1) = 1^2 + 1 = 1 + 1 = 2,$$

$$p_4(1) = 1 + 1 = 2,$$

$$p_5(1) = 1 - 1 = 0.$$

```
Adding these values together, we get:
```

$$p_1(1) + p_2(1) + p_3(1) + p_4(1) + p_5(1) = 5 + 1 + 2 + 2 + 0 = 10.$$

```
Thus, the final answer is:
```

$$\boxed{10}.$$

To qualitatively illustrate the Selection Dynamics of S2T-LOCAL, we present a representative reasoning trajectory.

**MMLU problem under S2T-LOCAL**

- Blue: S2T-LOCAL matches the LLM preference where the SLM's original top-1 was already correct.
- Red: S2T-LOCAL successfully follows the LLM to rescue the trajectory from the SLM's flawed ranking.
- Green: S2T-LOCAL fails to identify the LLM's preferred candidate, resulting in a selection error.

To translate the sentence "George borrows Hector's lawnmower" into predicate logic, we need to identify the subjects and objects involved in the action of borrowing.

1. **Identify the subjects and objects:** – George (g) is the subject performing the action. – Hector's lawnmower (l) is the object being borrowed. – The borrowing action is performed by George from Hector.

2. Understand the predicate: – The predicate ``$Bxyz$'' means ``x borrows y from z.''

3. **Apply the predicate to the sentence:** – In this case, George (g) is the subject (x), Hector's lawnmower (l) is the object (y), and Hector (h) is the source (z).

4. **Translate the sentence into predicate logic:** – George borrows Hector's lawnmower. – This translates to: $B_g(l, h)$.

5. **Match the translation with the given options:** – Let's check each option: – (A) $B_h(g, h)$ -- incorrect. – (B) $B_g(g, h)$ -- incorrect. – (C) $B_h(l, h)$ -- incorrect. – (D) $B_g(h, l)$ -- incorrect. – (E) $B_h(g, l)$ -- incorrect. – (F) $B_l(h, g)$ -- incorrect. – (G) $B_l(l, h)$ -- incorrect. – (H) $B_l(g, h)$ -- incorrect. – (I) $B_h(h, g)$ -- incorrect. – (J) $B_g(l, h)$ -- correct.

Therefore, the correct translation into predicate logic is:
The answer is (J).

