# OpenReview forum: "Select to Think: Unlocking SLM Potential with Local Sufficiency"
_ICML.cc/2026/Conference — ICML 2026 regular_

### Official Review · Reviewer_Co97 · 2026-02-13

**Soundness:** 3
**Presentation:** 3
**Significance:** 3
**Originality:** 3
**Overall Recommendation:** 5
**Confidence:** 4

**Summary:**

This work proposes S2T, based on the insight that the correct LLM-preferred token is often already within the SLM’s top-K predictions. By distilling this selection capability into the SLM, the method significantly improves reasoning performance while maintaining near-greedy inference efficiency without requiring LLMs at test time.

**Compliance With Llm Reviewing Policy:**

Affirmed.

**Final Justification:**

My concerns have been fully resolved. I thereby raise my score to 5.

**Key Questions For Authors:**

Please refer to the weakness.

**Limitations:**

I am good with no discussion of limitations.

**Strengths And Weaknesses:**

## Strength

1. This work reveals that the bottleneck in SLMs is not a lack of generalizability, but a misalignment in candidate ranking. The insight is potentially impactful.

2. S2T shows strong potential with minimal/light fine-tuning.

3. The experimental results are comprehensive and generally convincing, covering multiple benchmarks and settings to support the effectiveness of S2T.


## Weakness

1. The experiment is based on only Qwen2.5-Instruct family. It would be stronger if Llama-3.2 could be included.

2. In LLM-involved scenarios, S2T does not show a clear advantage over speculative decoding. I believe that S2T can be viewed as a form of speculative decoding with single-step verification.

3. I expect the authors to release a runnable code and trained checkpoints that are ready to execute, even for the reviewing process.

---

> ### Author Rebuttal · Authors · 2026-03-29
>
> __We sincerely thank the reviewer for the constructive feedback, and for recognizing the potential impact of our ranking-misalignment insight.__
>
> __Q1 on Generalization to Llama-3.2.__
>
> During the rebuttal period, we trained and evaluated the full S2T and S2T-LOCAL pipelines on a non-Qwen pair: __Llama-3.2-1B (student) and Llama-3.1-8B (teacher)__. The downstream results are:
>
> |Dataset|SLM Greedy (%)|S2T-LOCAL (%)|S2T (%)|
> |-|-|-|-|
> |GSM8K|37.5|43.6|56.7|
> |MATH500|27.8|34.6|45.5|
> |OlympiadBench|2.5|6.4|8.1|
> |MMLU-Pro|16.5|20.3|28.6|
>
> These results show that the gains transfer beyond Qwen: S2T provides a strong collaborative upper bound, giving S2T-LOCAL a rich supervision signal to learn from. Consequently, S2T-LOCAL consistently improves over the Llama greedy baseline in a fully teacher-free setting.
> Crucially, consistent with our Qwen setup, __this Llama S2T-LOCAL model was trained exclusively on the MATH dataset__. We further evaluated these exact checkpoints on general reasoning and open-ended generation (ARC-Challenge, TruthfulQA, MT-Bench; __see Reviewer sXE3, Q3__), yielding substantial gains without task-specific retraining.
> This demonstrates that the distilled selector captures the teacher's underlying reasoning preferences rather than overfitting to mathematical patterns.
>
> To further understand this transfer, we report the corresponding local-sufficiency and selector-alignment diagnostics. We track the tuned SLM’s greedy-decoding accuracy (FT Greedy Acc.) as a practical proxy for its retained reasoning performance.
>
> |Dataset|Hit@1 (%)|Hit@8 (%)|FT Greedy Acc. (%)|Agree@1 (%)|
> |-|-|-|-|-|
> |GSM8K|27.7|85.1|40.0|34.9|
> |MATH500|23.3|92.2|29.2|41.9|
> |OlympiadBench|27.2|87.9|4.8|33.6|
> |MMLU-Pro|21.7|86.4|17.5|29.6|
>
> The similar qualitative pattern is preserved on the Llama architecture: the large gap between Hit@1 and Hit@8 supports local sufficiency, and the distilled selector achieves useful teacher alignment (Agree@1) without obvious degradation in retained base-model behavior.
> In addition, we also validated local sufficiency and S2T performance on __Phi-3 and Gemma 2__; we report those broader results in our response to __Reviewer zZQf, Q1__.
>
> __Q2 on comparison with speculative decoding.__
>
> We agree that collaborative S2T has a resemblance to single-step speculative decoding, in that the SLM proposes and the LLM helps determine the next token.
>
> Rather than a weakness, the fact that S2T achieves competitive performance to open-ended generation under a much more restricted intervention is itself __the central finding of our work__. It suggests that, at critical reasoning states, bounded-support selection over SLM proposals is often sufficient, so the useful teacher signal can be reduced from full-vocabulary generation to a small candidate-set decision problem.
>
> This is __the key distinction from speculative decoding__. Speculative decoding is designed to accelerate LLM generation, but it does not change the underlying learning target. In contrast, S2T identifies a different intervention primitive: selection rather than generation. This reformulation bypasses full distribution reproduction, making the teacher's guidance significantly easier for capacity-constrained students to internalize, ultimately enabling S2T-LOCAL.
>
> __Q3 on Code and Checkpoints.__
>
> In the current submission, the supplementary material already includes __runnable scripts covering the core training and decoding pipelines for S2T and S2T-LOCAL__. We acknowledge that we did not include pre-trained checkpoints in the review package, which is a limitation of the current release.
>
> For reproducibility, we summarize the key S2T-LOCAL hyperparameters for Qwen2.5 1.5B here. The number of reserved bins $|\mathcal R| = 16$. The predefined bin vector $v$ is fixed (not learned) and uniformly spaced over $[0, 1]$; the bin softmax temperature is $0.7$. For data denoising, we filter out samples where the teacher's score margin is below $\delta=0.08$. For optimization, we use LoRA with $r=16$, $\alpha=32$, dropout of 0.05, and a stability regularizer $\lambda_{\text{reg}}=30$.
> We will consolidate these values into a clear hyperparameter table in the revised appendix.
>
> In addition, we report hyperparameter sweeps (e.g., varying $\lambda_{\text{reg}}$ and $|\mathcal R|$) in __Reviewer sXE3, Q1__.
> Upon acceptance and de-anonymization, we will publicly release all distilled checkpoints (including the newly added Llama-3.2 models) together with a polished GitHub repository.
>
> We hope this response clarifies your concerns.

---

> > ### Author Rebuttal · Reviewer_Co97 · 2026-04-02
> >
> > My concerns have been fully addressed. I thereby raise my score to 5.

---

> > > ### Author Response · Authors · 2026-04-02
> > >
> > > We are thrilled that our additional evaluations addressed your concerns, and we sincerely thank you for recommending our work with a score of 5 (Accept).
> > > We deeply appreciate your recognition of our core insight, light fine-tuning efficiency, and comprehensive evaluation.
> > > As promised, we will integrate the new experiments into the revision and release the pre-trained checkpoints to complement the runnable code already provided.

---

### Official Review · Reviewer_iCJm · 2026-03-07

**Soundness:** 3
**Presentation:** 3
**Significance:** 3
**Originality:** 3
**Overall Recommendation:** 4
**Confidence:** 3

**Summary:**

This paper studies whether small language models can benefit from large-model guidance through selection rather than open-ended token generation. The central empirical observation is local sufficiency: at reasoning divergence points, the larger model’s preferred next token often already lies within the small model’s top-K candidate set, with the paper reporting about 95% Hit@8 for a 1.5B-to-32B pairing. Building on this, the paper proposes SELECT TO THINK (S2T), where the larger model ranks the small model’s local candidates instead of generating a replacement token, and S2T-LOCAL, which distills this selection logic into the small model using reserved-token logits so that inference can remain teacher-free. The method is evaluated on six reasoning/code/knowledge benchmarks across 0.5B and 1.5B students, with the main claim that S2T-LOCAL substantially improves greedy decoding and approaches the effectiveness of 8-path self-consistency while retaining near single-trajectory efficiency.

**Compliance With Llm Reviewing Policy:**

Affirmed.

**Final Justification:**

Thank you for the detailed rebuttal. It addresses my main concerns. I remain supportive of the paper and continue to place it in the weak accept range.

**Key Questions For Authors:**

- Can you report per-benchmark standard deviations or confidence intervals for Table 1, and ideally indicate which improvements remain significant under that uncertainty?
- What happens in the selector disagreement cases? In other words, when S2T-LOCAL disagrees with the teacher’s preferred candidate, are those disagreements usually harmless alternatives, or do they correlate with downstream failures?
- Can you provide full downstream S2T and/or S2T-LOCAL results on a non-Qwen architecture pair, rather than only hit-rate visualizations?
Right now the cross-architecture evidence is promising but incomplete. If the downstream gains transfer to another family, I would view the originality and significance claims as notably stronger.
- How tightly matched are the training and tuning conditions for the distillation baselines, especially DistilQwen2.5 and TSD-KD, relative to S2T-LOCAL?

**Limitations:**

Partially. The paper includes an impact statement, but it is fairly brief and mostly emphasizes the positive case of more resource-efficient reasoning systems.

**Strengths And Weaknesses:**

Strength:
- Regarding the soundness, the paper’s main strength is that its evaluation is well aligned with its central hypothesis. It first tests the prerequisite phenomenon of local sufficiency through Hit@K analysis, then tests whether the selection signal is learnable through selector-alignment metrics, and finally measures downstream benchmark accuracy and latency. The experimental setup is also thoughtful: the paper evaluates on six benchmarks spanning math, code, and general reasoning; it trains S2T-LOCAL only on MATH and evaluates OOD on the remaining tasks; and it includes a particularly valuable Takeover baseline that uses the same trigger schedule and budget as S2T while replacing selection with generation, which helps isolate the contribution of the selection mechanism itself. The appendix also provides a useful causal story through the hit-failure/selection-failure decomposition and ablations on trigger metrics and candidate diversity.
- For presentation, I found the paper generally clear and well structured. The high-level story is easy to follow: problem motivation, local-sufficiency insight, S2T framework, S2T-LOCAL distillation, then experiments and ablations.
- For significance, I think the paper tackles a meaningful and timely problem. There is real practical value in improving SLM reasoning without keeping a large teacher in the inference loop, and the reported latency savings are substantial enough to matter for deployment.

Weaknesses:
- regarding the soundness, first, although the paper states that results are averaged over three random seeds, it does not report per-benchmark standard deviations, confidence intervals, or significance tests, which matters especially for small-gain settings such as AIME25. Second, the distilled selector only reaches about 67–72% Agree@1 on the reported 1.5B settings, and the paper does not analyze whether disagreement cases are mostly benign or whether they correlate with downstream failures. Third, the strongest generality claim is only partially supported: the cross-architecture evidence on Gemma is helpful, but it is visualization-based rather than a full downstream S2T/S2T-LOCAL evaluation. Finally, I think one part of the narrative should be narrowed: the collaborative S2T results are often competitive, but Table 1 does not show uniform parity with the strongest generative collaborative baselines on every benchmark.
- My main presentation weaknesses are that a few claims are broader than the tables fully justify, and some summary statistics are slightly too polished relative to the nuance in the per-benchmark results. In particular, the “24.1% average improvement” headline aggregates across heterogeneous tasks with very different baseline levels, and the collaborative “matches generative methods” wording is somewhat stronger than what Table 1 supports benchmark-by-benchmark.
- The main limitation on significance is that the long-term impact still depends on how broadly local sufficiency transfers. Right now, the strongest downstream evidence is within one main family setup, and the cross-family evidence is encouraging but incomplete. So I do think the work advances the area, but I see it as a strong subarea contribution rather than a field-shifting result at this stage.

Originality:
-   I view the contribution as novel but not radically new. The strongest original element is the empirical and conceptual reframing: the paper argues that, at critical reasoning steps, SLM failures are often ranking failures rather than outright knowledge-absence failures, and it operationalizes that claim through the local-sufficiency analysis. The move from generation-based intervention to bounded selection is also a meaningful design simplification, and the teacher-free reserved-token realization is practically useful. At the same time, the paper is not built from entirely new ingredients: the broader space already contains token-level routing, selective distillation, verifier/reranking ideas, and ZIP-style reserved-token prediction. So the originality comes from a strong empirical insight plus a thoughtful recombination into a cleaner paradigm, rather than from an entirely unprecedented algorithmic family.

---

> ### Author Rebuttal · Authors · 2026-03-30
>
> __We sincerely thank the reviewer for the constructive feedback, and for recognizing our conceptual reframing, practical significance, and thoughtful evaluation design.__
>
> __Q1 on standard deviations for Table 1.__
>
> To address this, we report the mean ± standard deviation per-benchmark across independent runs with different random seeds for the main Greedy and S2T-LOCAL results in Table 1 (OB for OlympiadBench):
>
> |Model|Method|GSM8K|MATH500|OB|AIME25|HumanEval|MMLU-Pro|
> |-|-|-|-|-|-|-|-|
> |**0.5B**|Greedy|44.5±1.4|30.4±1.6|5.8±0.6|0.0±0.0|18.9±1.6|11.3±0.3|
> ||S2T-LOCAL|61.6±1.2|46.8±0.8|8.5±1.4|0.0±0.0|26.7±2.2|18.7±1.2|
> |**1.5B**|Greedy|72.1±1.2|54.4±2.2|20.1±0.7|1.1±1.5|42.7±1.8|22.9±1.4|
> ||S2T-LOCAL|86.6±1.0|67.5±1.5|27.0±2.4|2.2±0.5|56.9±2.0|33.6±1.2|
>
> The observed run-to-run variances are generally small relative to the absolute gains of S2T-LOCAL over Greedy, suggesting that the main improvements reflect robust empirical trends rather than seed artifacts. We agree that very small-gain cases such as AIME25 should be interpreted more cautiously.
> We will update Table 1 in the revision to include these mean ± std values.
>
> __Q2 on selector disagreement cases.__
>
> The short answer is: not all disagreements are fatal, but they are indeed strongly correlated with downstream failure.
>
> Quantitative Analysis: On our MATH and GSM8K validation sets, we partition trajectories into an agree group (zero selector–teacher disagreements across trigger steps) and a disagree group (at least one disagreement). The agree group achieves 80.1% final accuracy, whereas the disagree group drops to 32.3% (based on 900 sampled cases). Furthermore, the total number of disagreement steps per trajectory is negatively correlated with final correctness ($r=-0.46$, $p=1.40 \times 10^{-44}$). This indicates that selector disagreement is a meaningful predictor of the residual performance gap between the distilled S2T-LOCAL and the S2T.
>
> Qualitative Inspection: Manual inspection shows that disagreement cases are a mixture. A meaningful portion consists of harmless stylistic/semantic alternatives (e.g., `So` instead of `Now`) or valid reordered reasoning steps. However, the harmful ones are typically execution-critical deviations, such as wrong operators (e.g., `+` instead of `-`) or incorrect numerical values (e.g., `5` vs. `3`).
>
> __Q3 on non-Qwen architectures.__
>
> We trained and evaluated the complete S2T and S2T-LOCAL pipelines on the __Llama 3 family__ (Llama-3.2-1B student, Llama-3.1-8B teacher).
> The following results show that the gains transfer beyond Qwen: S2T provides a strong collaborative upper bound, enabling S2T-LOCAL to significantly improve over the greedy baseline in a teacher-free setting.
>
> |Dataset|SLM Greedy (%)|S2T-LOCAL (%)|S2T (%)|
> |-|-|-|-|
> |GSM8K|37.5|43.6|56.7|
> |MATH500|27.8|34.6|45.5|
> |OB|2.5|6.4|8.1|
> |MMLU-Pro|16.5|20.3|28.6|
>
> For additional diagnostics on this Llama setup (including local sufficiency and selector-alignment metrics), please see __Reviewer Co97, Q1__. We also validated collaborative S2T on __Phi-3 (3.8B -> 14B) and Gemma 2 (2B -> 27B)__, where we observe the similar improvement (please see __Reviewer zZQf, Q1__).
>
> Crucially, consistent with our Qwen setup, this Llama S2T-LOCAL model was __trained exclusively on the MATH dataset__. We further evaluated these exact checkpoints on general reasoning and open-ended generation, yielding substantial gains without task-specific retraining. Detailed results are provided in our response to __Reviewer sXE3, Q3__. This demonstrates that __the distilled selector captures the teacher's underlying reasoning preferences__ rather than overfitting to mathematical patterns.
>
> __Q4 on baseline definition.__
>
> DistilQwen2.5 is an off-the-shelf, industrial-grade baseline. According to its technical report (arXiv:2504.15027), it is trained using a massive multi-agent pipeline with proprietary LLM data augmentation and white-box model fusion. Thus, rather than a strictly controlled baseline, it serves as a practical reference for state-of-the-art distilled performance.
>
> TSD-KD is a student-centric, on-policy KD framework that selectively leverages teacher supervision. It integrates three objectives: direct distillation, indirect preference ranking, and entropy regularization. In our reproduction, we used the same student/teacher pair, the same training dataset, and the same overall training volume, while keeping the optimization protocol aligned with the published TSD-KD setup. Thus, __the key difference is not data scale or teacher access, but the learning target__: TSD-KD distills toward the teacher’s full-vocabulary behavior, whereas S2T-LOCAL reformulates the problem as bounded-support candidate selection.
>
> __In addition__, we deeply value the presentation feedback. We will refine our phrasing in the revision to ensure all high-level claims are rigorously grounded in the specific per-benchmark performance.

---

> > ### Author Rebuttal · Reviewer_iCJm · 2026-04-01
> >
> > Thank you for the detailed rebuttal. The rebuttal resolves my main concerns and increases my confidence in a Weak Accept recommendation. Overall, I continue to recommend this paper within the weak accept range, as I see it as a solid and well-executed contribution with some remaining limitations in scope and validation.

---

> > > ### Author Response · Authors · 2026-04-02
> > >
> > > We thank the reviewer for the thoughtful follow-up and are glad the rebuttal resolved your main concerns, increasing your confidence in recommending our work. We deeply appreciate your recognition of our core conceptual reframing, its practical significance, and the strong alignment of our evaluation. We will incorporate all the supplementary analyses into the revision to further enrich the paper.

---

### Official Review · Reviewer_sXE3 · 2026-03-12

**Soundness:** 3
**Presentation:** 3
**Significance:** 3
**Originality:** 3
**Overall Recommendation:** 4
**Confidence:** 3

**Summary:**

The paper presents SELECT TO THINK (S2T), a framework that improves the reasoning capabilities of Small Language Models (SLMs) without the high inference costs of Large Language Models (LLMs). The authors identify "local sufficiency," observing that an LLM's preferred token is found within an SLM's top-8 predictions in 95% of divergence cases. S2T reframes LLM guidance from open-ended generation to a discrete selection task over the SLM's candidates. Furthermore, S2T-LOCAL distills this selection ability directly into the SLM using repurposed reserved tokens, allowing autonomous, teacher-free inference. The approach improves greedy decoding performance by an average of 24.1%.

**Compliance With Llm Reviewing Policy:**

Affirmed.

**Final Justification:**

The paper presents a solid contribution, with a few known limitations. While the initial submission contained some ambiguities regarding the implementation, the rebuttal successfully resolved these points. I recommend that the authors incorporate these clarifications into the final version of the manuscript. Overall, the rebuttal addressed my primary concerns and has reinforced my prior assessment of the work.

**Key Questions For Authors:**

1. Regarding the stability regularizer ($L_{reg}$​): You mention it requires a massive scaling coefficient ($\beta$ between 10-100) because the KL divergence term contributes a negligible fraction to the total loss. How sensitive is the model's linguistic coherence to the exact choice of $\beta$, and how was this tuned? Similarly, how sensitive is the distillation pipeline to the fixed size of 16 reserved tokens?

2. Could you provide the exact values or the initialization strategy for the predefined bin vector v used to compute the preference score in Equation 5?

3. Given that the framework mandates unrestricted sampling (top−p=1.0) to maintain candidate diversity, have you evaluated S2T-LOCAL on dedicated factuality and hallucination benchmarks to quantify whether this choice introduces regressions in open-ended generation?

4. How is the trigger mechanism dynamically thresholded during inference to hit specific intervention budgets (e.g., 1% or 20%) when it was trained specifically on a 10% KL divergence split?

**Limitations:**

- Bounded Recovery Capacity: The method fundamentally relies on the correct token existing within the SLM's top-K proposals; if the SLM completely fails to generate a viable path within that local neighborhood, the framework cannot recover.

- Complex Training Pipeline: Distilling the selector requires a highly specific, multi-phase training evolution to overcome bottlenecks like class imbalance and vocabulary interference.

**Strengths And Weaknesses:**

**Strengths**

- Novel Paradigm: Shifting from generative distribution matching to discrete candidate selection offers a highly efficient way to bypass the LLM-SLM capacity gap.

- Strong Empirical Validation: The "local sufficiency" hypothesis is rigorously validated across different scales (0.5B and 1.5B) and model families, proving to be a consistent phenomenon.

- Inference Efficiency: S2T-LOCAL drastically reduces latency, achieving a ~75% time reduction compared to collaborative LLM-takeover baselines.

- OOD Generalization: Despite training exclusively on the MATH dataset, the distilled selector successfully generalizes to out-of-distribution tasks like code generation (HumanEval) and general reasoning (MMLU-Pro).

**Weaknesses**

- Unablated hyperparameters: Critical hyperparameters, such as the 16 reserved tokens and the 0.08 teacher margin filter, are presented as hardcoded values without empirical justification or ablation studies.

- Missing Implementation Details: The paper omits critical reproducibility details, such as the exact numerical values for the predefined bin vector v used in the local scoring function.

- Trade-off Between Candidate Diversity and Factuality Controls: To achieve optimal ranking efficacy and avoid prematurely excluding LLM-preferred tokens, the S2T framework requires unrestricted sampling parameters, specifically setting top-p = 1.0. However, forcing unconstrained sampling removes the primary mechanism typically used to suppress hallucinations and maintain logical coherence. Because the paper's evaluation focuses strictly on reasoning accuracy (e.g., exact match math scores), it remains unclear whether mandating top-p = 1.0 severely degrades the model's performance in terms of factuality, grounding, or hallucination rates during open-ended generation tasks.

---

> ### Author Rebuttal · Authors · 2026-03-29
>
> __We sincerely thank the reviewer for recognizing our novel paradigm and strong empirical validation, and for the constructive feedback on implementation details.__
>
> __Q1 on hyperparameter sensitivity.__
>
> Below is a representative grid search of the stability regularizer $\lambda_{\text{reg}}$​ and the number of reserved bins $|\mathcal R|$ on Qwen2.5, 1.5B (MATH500).
> We track the tuned SLM’s greedy-decoding accuracy (FT Greedy) as a practical proxy for its retained reasoning performance.
>
> |$\mathcal{R}$|$\lambda_{\text{reg}}$ |S2T-LOCAL (%)|FT Greedy (%)|Agree@1 (%)|
> |-|-|-|-|-|
> |8|30|61.4|50.6|48.5|
> |12|30|69.8|55.1|66.9|
> |16|5|48.0|32.5|70.4|
> |16|30|67.5|49.6|67.1|
> |16|70|58.4|49.6|61.2|
>
> Two clear trends emerge.
> First, the large value of $\lambda_{\text{reg}}$ mainly reflects loss-scale alignment, since the raw KL term is much smaller than the selection loss. A too-small value (5) leads to degradation in base behavior, while a too-large value (70) over-constrains selector learning.
> Second, $|\mathcal{R}|=12, 16$ perform strongly, while $|\mathcal{R}|=8$ under-resolves the ranking score.
> Thus, we selected $|\mathcal{R}|=16$ and $\lambda_{\text{reg}}=30$ as the final configuration based on broader validation across tasks.
>
> __Q2 on the bin vector.__
>
> The predefined bin vector $v$ is fixed (not learned), and set as a uniformly spaced sequence across the $[0, 1]$ range. For $|\mathcal{R}|=16$ discrete bins, we use $v_i = \frac{i-1}{16-1}$, yielding the frozen scalar anchors $[0.0, 0.0667, \dots, 1.0]$. These anchors are used to map the predicted categorical distribution over bins back to a continuous preference score in Eq. 5.
>
> __Q3 on unrestricted sampling.__
>
> $p=1.0$ is used strictly at trigger points to preserve the SLM's full local support. Truncating this support often excludes teacher-preferred tokens and degrades performance (see Table A6).
> Moreover, this does not imply unconstrained stochastic token generation: the selector deterministically picks from the bounded candidate pool, while all non-trigger steps remain standard greedy decoding. Thus, the usual hallucination concerns do not apply in the same way here.
>
> To test for regressions in broader non-math domains, we evaluated __the exact same checkpoints trained only on MATH from our main paper__ on factuality, reasoning, and open-ended generation.
> Remarkably, S2T-LOCAL achieves broad average relative gains over greedy decoding on TruthfulQA (+38.9%), ARC-C (+23.8%), and MT-Bench (+3.3%):
>
> |Model|Benchmark|SLM Greedy (%)|S2T-LOCAL (%)|S2T (%)|
> |-|-|-|-|-|
> |**Qwen 2.5 (0.5B)**|TruthfulQA|18.2|34.2|54.3|
> ||ARC-C|29.9|42.5|77.1|
> ||MT-Bench|3.4|3.6|5.6|
> |**Qwen 2.5 (1.5B)**|TruthfulQA|46.7|54.6|68.6|
> ||ARC-C|61.0|73.9|88.6|
> ||MT-Bench|4.8|5.1|5.6|
> |**Llama 3.2 (1B)**|TruthfulQA|35.1|39.3|45.7|
> ||ARC-C|45.5|49.2|58.6|
> ||MT-Bench|4.3|4.2|4.8|
>
> These results suggest that preserving candidate diversity at trigger points does not introduce a factuality penalty; rather, our idea actively improves truthfulness, yielding clear gains in general domains.
> We will explicitly clarify this study’s scope regarding hallucinations in the revision.
>
> To justify our fixed $K=8$ configuration, we also evaluated dynamic candidate sizing (e.g., retaining tokens exceeding 5% of the top probability). We found that at critical decision points, the SLM is often miscalibrated, leading to premature pruning of teacher-preferred tokens. Consequently, this dynamic approach (avg $K \in [3.1, 3.6]$) underperforms the fixed $K=8$ in both coverage and accuracy:
>
> |Trigger rate|Avg k|Hit@k (%)|S2T Acc. (%)|
> |-|-|-|-|
> |1%|3.6|80.2|73.6|
> ||8|96.3|81.1|
> |2%|3.6|85.3|77.9|
> ||8|97.0|79.1|
> |10%|3.1|95.7|74.9|
> ||8|99.5|83.1|
>
> __Q4 on dynamic thresholding.__
>
> The 10% KL training split is used to curate “hard” examples for training the trigger head and is decoupled from the trigger budget used at inference time. To realize any target budget $\tau$ (e.g., 1% or 20%) at test time, we use a simple post-hoc percentile calibration: on a small held-out calibration set, we compute the trigger scores and set the threshold to the $(1-\tau)$-th percentile. This threshold is then applied during inference, reliably approximating the target budget despite natural distribution shifts.
>
> __L1 on bounded recovery.__
> We agree that it is an inherent limitation, and due to the inherent capacity gap, SLM's hit@K will not be 100%. Our point, however, is that the dominant bottleneck is selection failures rather than hit failures, and correcting the SLM’s flawed local ranking yields large gains even with a bounded candidate pool.
>
> __L2 on training complexity.__
> To avoid confusion, Appendix D.4's "Phases" record our design exploration (ablation of failed attempts), not a multi-phase training pipeline.
> Final selector training is __single-stage fine-tuning__: with a fixed predefined bin vector, training only adjusts reserved-token logits to match teacher's rankings, while KL regularization preserves normal generation.

---

> > ### Author Rebuttal · Reviewer_sXE3 · 2026-04-04
> >
> > I thank the authors for their thorough rebuttal, which has successfully addressed most of my concerns. The ablation study regarding the values of $R$ and $\lambda_{reg}$ is particularly interesting, especially given the accuracy's apparent sensitivity to these parameters. It would be valuable to investigate whether these optimal hyperparameter choices are universal across different model sizes and architectures.
> >
> > Overall, I will maintain my positive score for this paper.

---

> > > ### Author Response · Authors · 2026-04-04
> > >
> > > We sincerely thank the reviewer for the continued support and for raising this insightful question about hyperparameter transferability.
> > >
> > > During the rebuttal period, we trained a new S2T-LOCAL model on Llama-3.2 1B (see Reviewer Co97, Q1), using the same base configuration established from our Qwen2.5 sweeps.
> > > Our main finding is that __most key hyperparameters transferred well without retuning__ across the tested Qwen and Llama settings, including the bin temperature, LoRA setup, and stability regularizer $\lambda_{\text{reg}}$.
> > > The main parameter that required __a lightweight sweep__ was the reserved-bin count $|\mathcal{R}|$, which varies slightly depending on the specific model's architecture and capacity.
> > >
> > > Below, we provide the configurations under different model sizes and architectures evaluated on the MATH500 dataset for your reference:
> > >
> > > | Model | $\mathcal{R}$ | S2T-LOCAL (%) | FT Greedy (%) | Agree@1 (%) |
> > > |:---|:---|:---|:---|:---|
> > > | Qwen2.5 (1.5B) | 12 | 69.8 | 55.1 | 66.9 |
> > > | | 16 | 67.5 | 49.6 | 67.1 |
> > > | Qwen2.5 (0.5B) | 12 | 46.8 | 31.0 | 38.1 |
> > > | | 16 | 43.4 | 27.6 | 35.6 |
> > > | Llama-3.2 (1B) | 12 | 33.6 | 30.0 | 37.8 |
> > > | | 16 | 34.6 | 29.2 | 41.9 |
> > >
> > > Thus, our current takeaway is that the Qwen-derived configuration serves as a strong default starting point.
> > > We hope this cross-architecture validation thoroughly addresses your insightful query and further strengthens your positive recommendation for our work.

---

### Official Review · Reviewer_zZQf · 2026-03-13

**Soundness:** 2
**Presentation:** 3
**Significance:** 3
**Originality:** 2
**Overall Recommendation:** 4
**Confidence:** 3

**Summary:**

This article introduces two mechanisms that attempt to enhance the capacity of Small Language Models (SLM) to deal with reasoning tasks where they clearly underperform against LLMs. Thus, the paper proposes SELECT TO THINK (S2T), and S2T-LOCAL.

S2T is a collaborative reasoning paradigm designed to improve the reasoning quality SLMs without incurring the compute and latency costs of large language models (LLMs). The core empirical claim is that at points of reasoning divergence—where the SLM is likely to err—the LLM's preferred next token actually appears within the SLM’s top‑K logits with high probability (“local sufficiency”). Leveraging this property, the authors argue that the LLM can simply rank the SLM’s candidate tokens rather than generating them.

Moving further with this idea, the paper also introduces S2T‑LOCAL, a distillation method that teaches the SLM an internal ranking mechanism that reorders candidates using reserved-token logits, thereby eliminating the need for LLM calls at inference time.
Although S2T-LOCAL does not reach the performance of S2T, the paper provides empirical evidence that S2T‑LOCAL yields a 24.1% average accuracy improvement over greedy decoding across several reasoning benchmarks, and that a 1.5B SLM’s top‑8 candidates include the 32B LLM’s preferred token in 95% of divergence cases. The authors claim these gains allow S2T‑LOCAL to match the performance of 8‑path self-consistency while retaining single-path efficiency. If validated more broadly, the method could serve as a cost-efficient mechanism to enhance SLM reasoning.

**Compliance With Llm Reviewing Policy:**

Affirmed.

**Final Justification:**

The authors addressed in the rebuttal the main concerns I have regarding concepts (e.g. divergence points), clarifying questions about the code implementation as well as committing to improve the limitations section.

**Key Questions For Authors:**

# 1. How stable is the “95% hit@8” local sufficiency result across model families and tasks?
Your hit‑rate evidence comes primarily from Qwen‑2.5 models on math datasets. In the appendix Figure 2 you showed you tested it on Gemma-2-2B-IT and the hit@K were smaller (81%-84%). How about other SLMs like Llama, Phi, Pythia? Is it consistent across models and tasks?

# 2. What precisely defines a “divergence point,” and how consistent are these across runs?
The paper relies heavily on KL‑based divergence triggers but does not analyze: when are they calculated (at every token generation?), how stable these divergence points are across random seeds, whether different trigger metrics (entropy, margin, etc.) select similar steps, and whether the set of divergence points changes with minor model perturbations.
Strong evidence of stability would raise soundness. If divergence points vary widely, the core premise becomes less reliable and could lower the recommendation. There are some additional details in the appendix but not all these questions are addressed by the content of that sections.

# 3. Clarify the construction of candidate sets.
Specify precisely whether the LLM is only selecting from SLM top‑K or whether additional candidates (e.g., LLM’s top‑N) are ever injected. This is essential to validate the “local sufficiency” claim. In the python code shared, candidate‑set definition is (sometimes) not strictly “student‑only”: In the collaborative path, the code forms C by unioning SLM top‑K with LLM top‑N (C_ids = unique([slm_topk, llm_topn])). This departs from the paper’s strict claim that the LLM is only selecting among SLM proposals. Depending on how the experiments were run, this could inflate Hit@K and selection success. Clarification is needed.

# 4. What are the limitations of using reserved tokens for the ranking mechanism?
The use of reserved tokens to encode ranking logits presumes that the vocabulary includes unused token IDs and that the model architecture supports such repurposing. How robust is this across families where vocab and embeddings are tightly coupled or where resizing is constrained? Please provide failure cases and a fallback (e.g., adding auxiliary heads).

# 5. Discuss limitations and failure cases more thoroughly.
For example: Situations where local sufficiency does not hold, tasks requiring open-ended generation (such as creative writing), and risks of distilling biases in the teacher’s ranking preferences

**Limitations:**

Societal impact: I did not find an explicit discussion of potential negative impacts (e.g., deployment externalities from stronger small models, error modes in safety‑critical settings, or the effect on bias propagation when selection is learned from teacher biases). Adding a short, concrete section on this would be helpful.

**Strengths And Weaknesses:**

# Soundness

## Strengths
- The hypothesis of local sufficiency is well-motivated and experimentally validated with hit-rate statistics (e.g., 95% hit@8), suggesting genuine structure in SLM behavior.
- The shift from full distribution distillation to candidate ranking is logically sound and aligns with known difficulties of matching high-dimensional teacher distributions.

## Weaknesses
- **The notion of “divergence points” is central but underspecified**. The paper does not sufficiently analyze why these divergence points occur or whether they are stable across prompts, datasets, or sampling settings. I also miss more details on the cost of identifying these divergence points: does the method calculate the divergence at each new token generated?
- Experiments often focus on a SLM/LLM pair (1.5B vs 32B Qwen). It is unclear whether the 95% hit@8 holds for other architectures or more linguistically challenging tasks outside reasoning.
- The improvements (e.g., 24.1%) are large, but reliance on KL-triggering, calibrated thresholds, and top‑K selection adds procedural fragility. Robustness analyses (e.g., varying τ, K, alternative triggers) are not sufficiently thorough.
- **The theoretical section is lightweight and does not provide guarantees**; it mostly restates the decomposition of error without giving deeper insights.

# Presentation

## Strengths
- High-level narrative is clear: identify local sufficiency, then reframe LLM role, finally distill selection into the SLM.
- Figures effectively convey the comparison between generation and selection paradigms, specially Figure 1.
- Appendix is very complete and clarifies several aspects missing in the main text

## Weaknesses
- Captions sometimes require additional details, such as Figure 2, since the results in plots do not describe whether the model used was 0.5B or 1.5B parameters.
- Not enough clarity about model selection, training hyperparameters, or failure cases.
- Baselines such as TSD-KD or Takeover deserve clearer definitions—they are mentioned but not described enough for a reader to reproduce them without guessing.

# Significance

## Strengths
- If validated across broader settings, the idea of using SLM proposals as the only candidate space and replacing generation with selection could be highly influential for efficient model deployment.
- Matching 8-path self-consistency with single-pass inference is a meaningful achievement for practical efficiency.
- The focus on SLM reasoning performance is relevant as many institutions aim to deploy small, low-cost models.

## Weaknesses
- Impact may be overstated due to limited evaluation breadth, in terms of missing models and tasks
- Strong empirical gains are shown primarily on math reasoning; generalization to long-form tasks, safety-critical reasoning, or multilingual use is not demonstrated.
- Real-world deployment implications (latency, memory, batching) are not thoroughly discussed, for instance, how much cost does the trigger have if it needs to be executed after generating each token.

# Originality

## Strengths

- Clear conceptual novelty: treating LLM assistance as selection instead of generation is an interesting reframing rarely explored in previous distillation or collaborative inference works.
- The idea of distilling selection into a local critic aligns with, but does not duplicate, existing techniques.

## Weaknesses
- Despite the strengths in originality, assumptions are potentially fragile: The approach depends on a particular candidate set formation and the existence of reserved tokens for re-ranking. Implementation feasibility and robustness are not fully investigated or discussed.

---

> ### Author Rebuttal · Authors · 2026-03-29
>
> __We sincerely thank the reviewer for recognizing our conceptual novelty, and for the constructive feedback.__
>
> __Q1 on robustness of “Local sufficiency”.__
>
> We evaluated on three additional paired families (Llama 3.x, Phi-3, and Gemma 2) with our default trigger setting (1% KL-divergence) across both mathematical (MATH500) and general reasoning (MMLU-Pro):
>
> |Model|Dataset|SLM Greedy (%)|S2T Acc. (%)|Hit@1 (%)|Hit@8 (%)|
> |-|-|-|-|-|-|
> |**Llama 3.x (1B -> 8B)**|MATH|27.8|45.5|23.3|92.2|
> ||MMLU|16.5|28.6|21.7|86.4|
> |**Phi-3 (3.8B -> 14B)**|MATH|40.9|54.6|31.4|97.5|
> ||MMLU|27.1|41.3|26.3|93.5|
> |**Gemma 2 (2B -> 27B)** | MATH|31.2|40.4|9.9|84.9|
> ||MMLU|14.1|23.2|6.7|83.9|
>
> Results show that the large Hit@1-to-Hit@8 gap is not specific to Qwen. While absolute Hit@8 level varies by family, the relative surge is consistent: even in the most challenging case (Gemma 2, 2B), Hit@8 still reaches ~84%.
> We have also evaluated __general-domain tasks (ARC-Challenge, TruthfulQA, MT-Bench; see Reviewer sXE3, Q3)__, and trained __a full S2T-LOCAL model on Llama-3.2-1B (see Reviewer Co97, Q1)__, both validating our claims.
>
> These trends collectively support our core insight: even when SLM fails at its greedy prediction, the teacher-preferred token often remains within its top-8 local support.
>
> __Q2 on divergence points.__
>
> (1) Definition and calibration. A divergence point is a decoding step whose trigger score exceeds a calibrated threshold, e.g., the token-level KL divergence between LLM and SLM. The score is computed at every decoding step, and the threshold is set post-hoc on a small calibration set to match the target intervention budget. For deployment, S2T-LOCAL avoids runtime LLM dependence by using a lightweight MLP over the SLM hidden states to directly predict divergence scores.
>
> (2) Trigger robustness. Across minor perturbations, exact token-level divergence points naturally shift, since autoregressive paths change. Our claim is that the Local Sufficiency phenomenon remains robust even when the precise intervention locations shift.
> To test it, we compared very different metrics at a fixed 1% budget (MATH, Qwen2.5, 1.5B -> 32B):
>
> |Trigger Metric|S2T Acc. (%)|Hit@1 (%)|Hit@8 (%)|
> |-|-|-|-|
> |KLD|81.1|32.9|96.3|
> |SLM Entropy|56.5|40.0|95.8|
> |Random|55.5|93.4|100.0|
>
> These metrics select drastically different intervention steps, as evidenced by S2T Accuracy dropping from 81.1% to ~56%. Crucially, however, Hit@8 remains consistently high across all three triggers, suggesting that Local Sufficiency is not specific to the exact trigger locations.
>
> __Q3 on candidate construction.__
>
> We confirm that __all reported S2T / S2T-LOCAL results use strictly student-only candidate sets__. No LLM-derived candidates are ever injected into the generation path.
>
> The confusion stems from an interleaved logging/oracle-analysis block containing `unique([slm_topk, llm_topn])`. This union array is strictly for on-the-fly metric tracking and is never used in the actual S2T/S2T-LOCAL decoding.
> To verify this in the provided code:
> * Generation path: `slm_topk_ids` are extracted solely from the raw SLM logits (Lines 705, 711), and the candidate set is then constructed from `slm_topk_ids` (Lines 782–784). The subsequent mapping step (Lines 785-786) merely maps sampled local indices back to the global token IDs, without introducing any LLM-derived tokens.
> * Hit@K calculation: our reported Hit@K is computed by ranking the LLM's top token directly against the raw SLM logits (Lines 943, 949), entirely independent of the union array.
>
> We apologize that mixing tracking and decoding logic in the code caused this valid concern; we will separate these paths clearly in the final release.
>
> __Q4 on reserved tokens.__
>
> We agree that repurposing token IDs is not universally applicable. A concrete failure case is Llama 2, which does not publicly expose an obvious bank of reusable reserved slots, unlike newer families such as Llama 3 or Phi-3.
> Importantly, reserved tokens are a convenient implementation, not a conceptual requirement of S2T-LOCAL. As the reviewer suggests, the natural fallback is a lightweight auxiliary head parallel to the LM head. This preserves S2T's core generation-selection decoupling with negligible overhead.
>
> __Q5 on limitations.__
>
> We will expand the limitations section. (1) Open-ended generation: see Reviewer sXE3, Q3. (2) Sufficiency failure: see Reviewer sXE3, L1. (3) Bias transfer: distilling teacher preferences can propagate its biases, making fairness-sensitive auditing necessary.
>
> __Weakness.__
>
> Regarding baseline definition, Takeover baseline is a collaborative ablation: it shares S2T’s exact trigger schedule and budget, while replacing selection with LLM generation, which helps isolate the contribution of the selection mechanism itself; see Reviewer iCJm, Q4 for others.
>
> For additional details, please see Reviewer sXE3 (Q1, Q3) for hyperparameter and candidate size analysis, and Reviewer iCJm (Q2) for selector failure cases.

---

> > ### Author Rebuttal · Reviewer_zZQf · 2026-04-02
> >
> > The main concerns I pointed out are resolved, I appreciate the authors running additional experiments and clarifying aspects related to code implementation, as well as acknowledging the need for enriching the limitations section. I raised my score accordingly.

---

> > > ### Author Response · Authors · 2026-04-02
> > >
> > > We are glad that our new experiments and clarifications successfully resolved your main concerns, and we sincerely thank you for your positive recommendation.
> > > We are highly encouraged that you found our approach a "clear conceptual novelty" that reveals "genuine structure" and could be "highly influential."
> > > We will carefully integrate all these updates and clarifications into the revision.

---

### Decision · Program_Chairs · 2026-04-30

**Decision:**

Accept (regular)

**Comment:**

All reviewers agree that the paper is technically solid with well-written motivation (from generation to selection) and detailed explanation. Authors also responded with thorough additional results (e.g., different architectures). The method also has practical benefits of not using LLM in the deployment, which is nice. AC thinks this idea is interesting, somehow aligned with the popular approach of online policy distillation, which may be why it works reasonably well. As a bonus point, AC encourages the authors to try this idea on larger models.